# Cancer cell-expressed BTNL2 facilitates tumour immune escape via engagement with IL-17A-producing γδ T cells

Yanyun Du[1,10], Qianwen Peng[1,10], Du Cheng[2], Ting Pan[1], Wanwei Sun[1], Heping Wang[1], Xiaojian Ma[1], Ruirui He[1,3,4], Huazhi Zhang[1], Zhihui Cui[1], Xiong Feng[1], Zhiqiang Liu[1], Tianxin Zhao[1], Wenjun Hu [1], Lei Shen[2], Wenyang Jiang[5], Na Gao[6], Bradley N. Martin [7], Cun-Jin Zhang [8], Zhanguo Zhang[9] & Chenhui Wang [1,3,4✉]

Therapeutic blockade of the immune checkpoint proteins programmed cell death protein 1 (PD-1) and cytotoxic T lymphocyte antigen 4 (CTLA4) has transformed cancer treatment. However, the overall response rate to these treatments is low, suggesting that immune checkpoint activation is not the only mechanism leading to dysfunctional anti-tumour immunity. Here we show that butyrophilin-like protein 2 (BTNL2) is a potent suppressor of the anti-tumour immune response. Antibody-mediated blockade of BTNL2 attenuates tumour progression in multiple in vivo murine tumour models, resulting in prolonged survival of tumour-bearing mice. Mechanistically, BTNL2 interacts with local γδ T cell populations to promote IL-17A production in the tumour microenvironment. Inhibition of BTNL2 reduces the number of tumour-infiltrating IL-17A-producing γδ T cells and myeloid-derived suppressor cells, while facilitating cytotoxic CD8+ T cell accumulation. Furthermore, we find high BTNL2 expression in several human tumour samples from highly prevalent cancer types, which negatively correlates with overall patient survival. Thus, our results suggest that BTNL2 is a negative regulator of anti-tumour immunity and a potential target for cancer immunotherapy.

[1] Key Laboratory of Molecular Biophysics of the Ministry of Education, National Engineering Research Center for Nanomedicine, College of Life Science and Technology, Huazhong University of Science and Technology, 430074 Wuhan, China. [2] Department of Gastroenterology, Renmin Hospital of Wuhan University, Wuhan, China. [3] The Key Laboratory for Human Disease Gene Study of Sichuan Province and the Department of Laboratory Medicine, Sichuan Provincial People's Hospital, University of Electronic Science and Technology of China, 611731 Chengdu, China. [4] Research Unit for Blindness Prevention of the Chinese Academy of Medical Sciences (2019RU026), Sichuan Academy of Medical Sciences and Sichuan Provincial People's Hospital, Chengdu, Sichuan, China. [5] Department of Thoracic Surgery, Renmin Hospital of Wuhan University, Wuhan, China. [6] Key Laboratory of Molecular Biophysics of the Ministry of Education, Hubei Key Laboratory of Bioinformatics and Molecular-imaging, Department of Bioinformatics and Systems Biology, College of Life Science and Technology, Huazhong University of Science and Technology, 430074 Wuhan, China. [7] Division of pulmonary and critical care medicine, Brigham and Women's Hospital, Harvard Medical School, Boston, MA, USA. [8] Department of Neurology of Drum Tower Hospital, Medical School and the State Key Laboratory of Pharmaceutical Biotechnology, Nanjing University, Nanjing, Jiangsu, China. [9] Department of Hepatic Surgery Center, Tongji Hospital, Tongji Medical College, Huazhong University of Science and Technology, 430070 Wuhan, China. [10] These authors contributed equally: Yanyun Du, Qianwen Peng. ✉email: wangchenhui@hust.edu.cn

A significant recent advance in cancer immunology has been the discovery of immune resistance mechanisms in the tumour microenvironment (TME), which facilitate tumour escape from immunosurveillance. For example, it is known that IFN-γ, produced by tumour-infiltrating T cells, enhances the expression of immune checkpoint molecules such as PD-L1 (CD274, B7-H1), which in turn engage cognate receptors on effector T cells to promote exhaustion and apoptosis[1–4]. Antibody-mediated blockade of PD-1/PD-L1 and/or CTLA-4 immune checkpoint molecules reverses T cell dysfunction and restores effective anti-tumour immune responses[5,6]. This strategy, commonly referred to as immune checkpoint blockade (ICB), has been approved for the treatment of multiple cancer types[7–10]. Despite the dramatic results seen in some patients in response to currently available ICB therapy, the overall response rate remains disappointingly low[5,6,9,11]. Studies have shown that the PD-1/PD-L1 pathway is responsible for less than 40% of the immune dysfunction observed in human solid tumours, and accumulating evidence suggests that other mechanisms contribute to dysfunctional anti-tumour immunity in the TME[12–16]. Therefore, the elucidation of additional mechanisms of cancer immune evasion will both inform our understanding of ICB treatment response, while also provide novel targets to improve immunotherapeutic approaches for cancer.

γδ T cells are a non-major histocompatibility complex (MHC)-restricted lymphocyte subset closely aligned with innate immunity. γδ T cells preferentially localize within epithelial-rich tissues, such as the intestinal tract, skin and lungs. γδ T cells recognize MHC-independent nonpeptide antigens expressed during the cell stress response. γδ T cells have also been shown to execute anti-tumour-cytotoxic responses via multiple effector mechanisms, including the production and release of IFN-γ, as well as perforin and granzyme family members[17]. However, in certain contexts γδ T cells have also been shown capable of promoting tumour growth via the production IL-17 A. In fact, multiple recent reports have described tumour-promoting roles for γδT17 cells, both in murine models and in human cancer[18–23]. In these studies, myeloid-derived suppressor cells (MDSCs) were shown to be a key downstream target of tumour-promoting γδT17 cells. MDSCs are a heterogeneous population of suppressive innate immune cells that expand in the context of several disease states, including cancer. Several reports have found the MDSC population substantially increased in tumours of patients harbouring a broad array of cancers, including colon cancer and melanoma[23,24]. Furthermore, high levels of circulating MDSCs are predictive of poor response to ICB therapy, suggesting that MDSCs may play a role in immune escape[25,26]. Although MDSCs were known to play a clear promoting role for multiple types of cancers, there are currently no approved therapeutic agents that specifically target MDSCs.

Butyrophilin-like protein 2 (BTNL2) is a transmembrane immunoregulatory protein that is highly expressed in the gastrointestinal tract. BTNL2 belongs to the butyrophilin-like family of proteins, and several of the butyrophilins and butyrophilin-like proteins, such as BTN2A1, BTN3A1, skint-1, BTNL1 and BTNL6, have been shown to play an essential role in regulating γδ T cell development and differentiation[27–34]. BTNL2 contains four extracellular immunoglobulin (Ig) domains, comprised of two IgV-IgC domain pairs[35]. Previous studies reported that a BTNL2-Fc fusion protein directly inhibited CD4$^+$ T cell activation in vitro, although the cognate receptor for BTNL2 on T cells has remained unknown[36,37]. Intriguingly, it has also been reported that polymorphic variants of BTNL2 are associated with susceptibility to several autoimmune diseases and cancer, including lung and prostate adenocarcinoma[38–47]. One recent clinical study found that the expression of BTNL2 and other immune checkpoint molecules, including CTLA-4, was increased following anti-PD-1 therapy, further suggesting that BTNL2 may represent a novel mechanism of cancer immune evasion[48].

In this study, we provide evidence that BTNL2 inhibits anti-tumour immunity by acting on local γδ T cell populations to promote IL-17A production, which enhances tumour immune resistance via recruitment of MDSCs. Blockage of BTNL2 with a novel monoclonal neutralizing antibody has a significant therapeutic effect for multiple mice tumours, and has a synergistic effect with anti-PD-1 blockage. Importantly, BTNL2 is expressed in multiple human solid cancers, and its expression level is negatively correlated with patient survival. In summary, we report that BTNL2 is a promising cancer immunotherapeutic target, with potential to enhance the efficacy of currently available immunotherapies, and possibly also to offer alternative stand-alone therapy in patients who are resistant to the current cancer immunotherapy.

## Results

**BTNL2 blockade has a therapeutic anti-tumour effect in multiple murine models.** Given the compelling human genetic and translational data in the literature, we hypothesized that antibody-mediated blockade of BTNL2 would enhance the anti-tumour immune response. To examine this, we generated a library of rat monoclonal antibodies directed against murine BTNL2, and conducted functional in vitro screening that identified a single clone (mAb-2, referred to as BTNL2 mAb) capable of rescuing the BTNL2-Fc-mediated inhibition of CD4$^+$ T cells (Supplementary Fig. 1a). To further validate this antibody, we made BTNL2-KO mice by crispr-cas9, and deleted exon 1–3 of BTNL2 gene (Supplementary Fig. 1b). We found that BTNL2 mAb-2 antibody can be used for western blot and flow cytometry analysis (Supplementary Fig. 1c–e). As BTNL2 mAb-2 also recognized a non-specific protein of 95 kD, we performed the experiment of membrane-cytoplasm isolation, and found that this non-specific protein of 95 kD was exclusively expressed in the cytoplasm (Supplementary Fig. 1d). This data also suggest that this non-specific protein will not affect the in vivo function of mAb-2. Strikingly, both intraperitoneal (i.p.) and intravenous (i.v.) delivery of anti-BTNL2 mAb to tumour-bearing mice significantly reduced Lewis lung cancer (LLC) tumour growth (Fig. 1a). A similar anti-tumour effect was also found following i.p. injection of BTNL2 mAb into CT26 (murine colonic adenocarcinoma) or A20 (murine B cell cancer) tumour-bearing mice (Fig. 1b, c). In the A20 tumour model, five out of seven mice demonstrated a complete response with regression of all macroscopic tumour burden following anti-BTNL2 mAb treatment. We then re-implanted A20 tumour in the contralateral flank of these mice and failed to detect any evidence of tumour engraftment after re-implantation, indicating the development of long-term anti-tumour immune memory in response to treatment with anti-BTNL2 mAb (Fig. 1d). Following intravenous delivery of A20 tumour cells, a model of widely metastatic disease burden, anti-BTNL2 mAb substantially prolonged the survival of tumour-bearing mice (Fig. 1e). Next, we found that combinational treatment with anti-BTNL2 mAb and anti-PD-1 mAb had an additive anti-tumour effect compared to a single treatment, and anti-BTNL2 mAb and anti-PD-1 mAb had similar anti-tumour effects (Fig. 1f–h).

Next, we assessed the expression of BTNL2 in the mouse tumour microenvironment (TME). Interestingly, we observed a protein band at ~55 kDa that was strongly induced in engrafted LLC tumours compared to LLC cells, while the 72 kDa band was unchanged following engraftment. Expression of BTNL2 mRNA was significantly induced in engrafted LLC and CT26 tumours

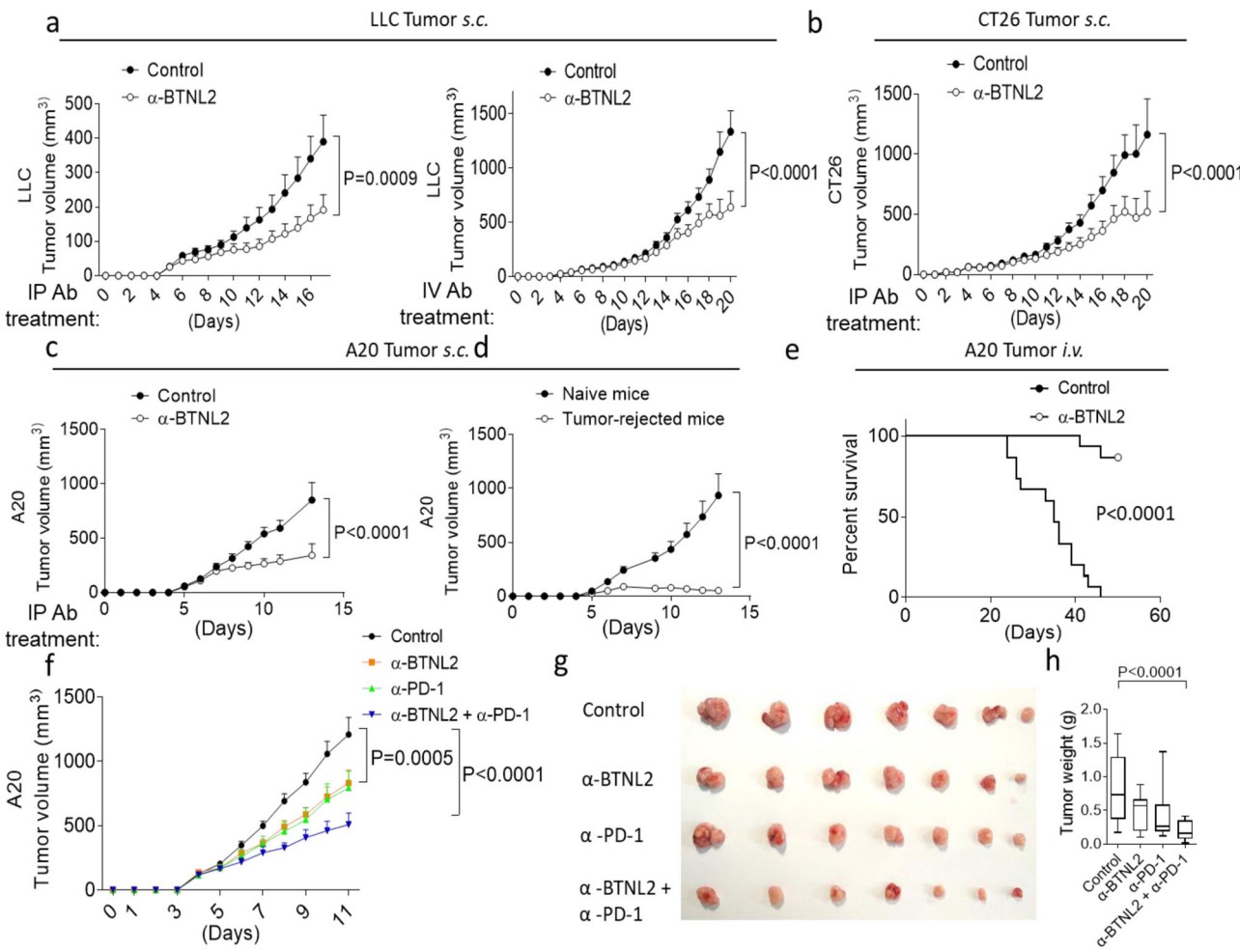

**Fig. 1 Anti-BTNL2 mAb has therapeutic effect for multiple tumours. a** Primary LLC tumour growth kinetics of mice after intraperitoneal injection of isotype rat IgG1 control Ab or anti-BTNL2 mAb (200 µg/mouse) (left panel) or intravenous injected of antibody (200 µg/mouse) (right panel) was shown. ($n = 13$, $P = 0.0009$ for left panel, and $n = 14$, $P < 0.0001$ for right panel). **b**, **c** Primary CT26 (**b**, $n = 14$ for each group, $P < 0.0001$) or A20 (**c**, $n = 17$ for each group, $P < 0.0001$) tumour growth kinetics of mice after intraperitoneal injection of antibody (200 µg/mouse) was shown. **d** Tumour free mice from anti-BTNL2 mAb treated group in **c** were re-implanted A20 tumours in the contralateral flank of mice, and tumour growth kinetics of mice was shown ($n = 12$ for each group, $P < 0.0001$). **e** Mice were intravenous injected $2 \times 10^6$ A20 tumour cells, followed by intraperitoneal injection of isotype control Ab or anti-BTNL2 mAb as described in the Materials and methods ($n = 15$ for each group, $P < 0.0001$) (200 µg/mouse). Mice survival was shown. **f** Primary A20 tumour growth kinetics of mice after intraperitoneal injection of control Ab, anti-BTNL2 mAb, anti-PD-1 mAb or anti-PD-1 mAb plus anti-BTNL2 mAb was shown. (200 µg/mouse of anti-BTNL2 mAb and 100 µg/mouse of anti-PD-1 mAb) ($n = 13$ for each group, $P = 0.0005$ for Control vs α-BTNL2, $P < 0.0001$ for Control vs α-BTNL2 + α-PD-1, $4 \times 10^6$ A20 cells were subcutaneously injected). **g** Tumour image from **f** was shown. **h** Tumour weight was shown ($n = 13$ for each group, $P < 0.0001$ for Control vs α-BTNL2 + α-PD-1, $4 \times 10^6$ A20 cells were subcutaneously injected). All data are mean ± s.e.m. *$P < 0.05$, **$P < 0.01$, ***$P < 0.001$, ****$P < 0.0001$ based on Two-way ANOVA for (**a–d**, **f**), Log-rank (Mantel-Cox) Test for (**e**) and one-way ANOVA for (**h**). Data are representative of three independent experiments (**a–e**) and two independent experiments (**f–h**).

compared to primary tumour cells, which was similar to *PD-L1* mRNA induction (Supplementary Fig. 2a,b). Notably, *BTNL2* mRNA induction was much greater than *PD-L1* induction in LLC tumours, which may explain at least in part the significant impact of anti-BTNL2 mAb treatment on LLC tumour growth relative to anti-PD-1 mAb treatment (Supplementary Fig. 2c). Interestingly, BTNL2 protein expression was significantly increased in LLC tumours after anti-PD-1 mAb treatment (Supplementary Fig. 2d), mirroring the prior report in humans that BTNL2 expression was upregulated following anti-PD-1 treatment[48]. After treatment with glycosylation inhibitor PNGase F, the intensity of the 72 kDa BTNL2 band decreased while the previously observed 55 kDa BTNL2 band appeared, indicating that the 55 kDa BTNL2 band represents the native non-glycosylated form of BTNL2 (Supplementary Fig. 2e). Following site-directed mutagenesis of four predicted glycosylation sites on BTNL2 (N210S, N296S, N427S

and N432S), we observed a return to the predicted molecular weight by SDS-PAGE, which indicates that BTNL2 is glycosylated at these four sites (Supplementary Fig. 2f). Flow cytometric analysis of the TME indicated that BTNL2 was primarily expressed on CD45− tumour cells; however, 48.47% of CD45+ leukocytes did also express BTNL2 (Supplementary Fig. 2g).

**BTNL2 inhibition reduces tumour-infiltrating γδT17.** BTNL2 belongs to the butyrophilin-like family of proteins, and many of the butyrophilins and butyrophilin-like proteins have been shown to play an essential role in the regulation of γδ T cell development and differentiation[27–34]. Interestingly, infiltration of CT26 and A20 tumours by γδT17 as well as serum IL-17A concentration were all greatly reduced following anti-BTNL2 mAb treatment compared to treatment with isotype control antibody (Fig. 2a, b). We also performed intravenous injection of MC38 and CT26

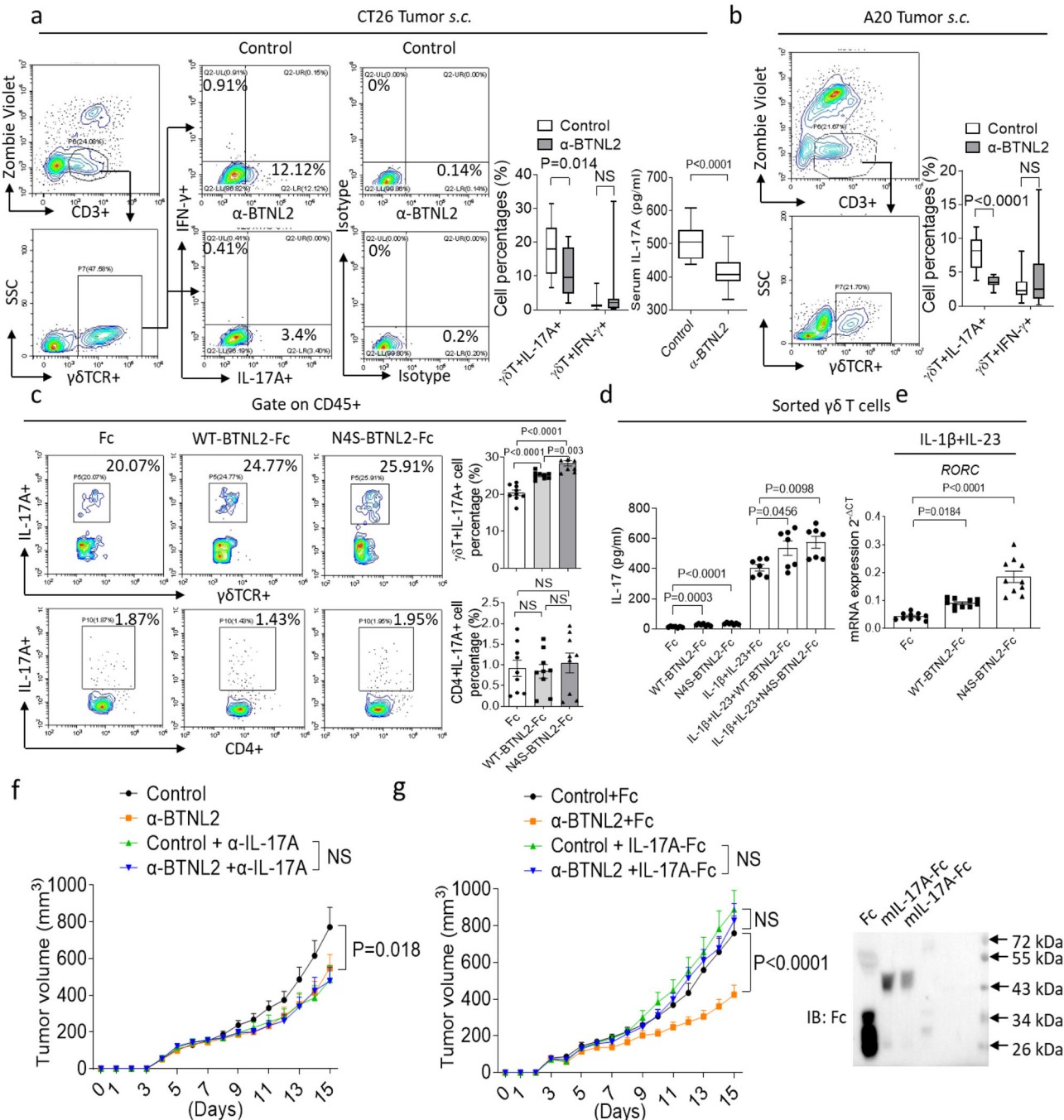

cells, and found that survival of tumour-bearing mice was significantly extended by anti-BTNL2 mAb treatment compared to controls (Supplementary Fig. 3a–d). It was reported that there were different types of immune cells infiltrated in the CT26 tumour, and we examined whether there was any difference in terms of other immune cell populations infiltration in the CT26 tumour after anti-BTNL2 mAb blockage[49,50]. Notably, there was no significant difference in the number of tumour-infiltrating IL-17A- or IFN-γ-producing CD4+ T cells, Treg cells or NK cells (Supplementary Fig. 3e–g). Next, we found that both wild-type (WT) BTNL2-Fc and N4S-BTNL2-Fc recombinant proteins were capable of inducing the production of IL-17A by γδ T cells in splenocytes (Supplementary Fig. 4a and Fig. 2c), and this effect was enhanced by IL-1β and IL-23 co-treatment (Fig. 2d)[51]. Notably, the glycosylation site mutant N4S-BTNL2-Fc promoted

significantly greater numbers of γδT17 and production of IL-17A than did WT-BTNL2, either with or without co-stimulation with IL-1β and IL-23 (Fig. 2c, d). Consistent with this, N4S-BTNL2-Fc promoted greater gene expression of the IL-17A transcriptional factor *RORC* than did WT-BTNL2-Fc (Fig. 2e). This was consistent with the finding that non-glycosylated BTNL2 was mainly induced in the TME (Supplementary Fig. 2a).

To further examine the role of IL-17A in tumour regression following BTNL2 blockade, we neutralized IL-17A in tumour-bearing mice and found that this completely abolished the anti-tumour effect of BTNL2 blockade (Fig. 2f). To further confirm the important role of IL-17A in the BTNL2 blockade, we purified murine IL-17A-Fc recombinant proteins (Fig. 2g, right panel), and found that treatment with recombinant IL-17A-Fc also abolished the anti-tumour effect of BTNL2 blockade (Fig. 2g, left

**Fig. 2 Anti-BTNL2 mAb treatment decreases the tumour infiltration of γδT17 cells. a, b** After isotype control Ab or anti-BTNL2 mAb treatment (200 μg/mouse), infiltrated live CD3 + γδ T lymphocytes which producing IL-17A or IFN-γ in subcutaneous CT26 (**a**) and A20 (**b**) tumours were analyzed by flow cytometry as indicated (**a**, n = 15, P = 0.014 for γδT+IL-17A+ cell Percentages, NS for γδT+IFN-γ+ cell Percentages and **b**, n = 14, P < 0.0001 for γδT+IL-17A+ cell Percentages, NS for γδT+IFN-γ+ cell Percentages). Serum IL-17A was examined by ELISA (n = 15, P < 0.0001 for **a**). **c** Splenocytes were cultured in the presence of plate-coated Fc, WT-BTNL2-Fc or N4S-BTNL2-Fc recombinant proteins (10 μg/ml) for 48 h, followed by flow cytometry analysis of γδT17 and Th17 (cells were restimulated with Cell Activation Cocktail (with Brefeldin A) for 4 h, and were gated by live CD45+, n = 9, P < 0.0001 for Fc vs WT-BTNL2-Fc γδT+IL-17A+ cell Percentages, P < 0.0001 for Fc vs N4S-BTNL2-Fc γδT+IL-17A+ cell Percentages, P = 0.003 for WT-BTNL2-Fc vs N4S-BTNL2-Fc γδT+IL-17A+ cell Percentages, NS for CD4+IL-17A+ cell Percentages). **d** FACS sorted γδ T cells were cultured in the presence of plate-coated Fc, WT-BTNL2-Fc or N4S-BTNL2-Fc with or without IL-1β and IL-23 for 24 h. ELISA was performed to analyze IL-17A production (n = 7, P = 0.0003 for Fc vs WT-BTNL2-Fc, P < 0.0001 for Fc vs N4S-BTNL2-Fc, P = 0.0456 for IL-1β + IL-23+Fc vs IL-1β + IL-23+WT-BTNL2-Fc, P = 0.0098 for IL-1β + IL-23+Fc vs IL-1β + IL-23 + N4S-BTNL2-Fc). **e** FACS sorted γδ T cells were cultured in the presence of plate-coated Fc, WT-BTNL2-Fc or N4S-BTNL2-Fc together with IL-1β and IL-23 for 24 h, followed by real-time PCR analysis of *RORC* expression (n = 10, P = 0.0184 for Fc vs WT-BTNL2-Fc, P < 0.0001 for Fc vs N4S-BTNL2-Fc). **f** IL-17A were neutralized by neutralizing antibody described in the Methods (100 μg/mouse), and CT26 tumour growth kinetics was shown (n = 16 for each group, P = 0.018). **g** Primary CT26 tumour growth kinetics of mice after intraperitoneal injection of control Ab or anti-BTNL2 mAb (200 μg/mouse) together with Fc or IL-17A-Fc recombinant proteins was shown. (n = 15 for each group, P < 0.0001). Fc or IL-17A-Fc recombinant proteins (5 μg/mouse) were intraperitoneal injected at day 1, 4, 7, 10 and 13 after tumour implantation. Right panel indicates the purified Fc and IL-17A-Fc recombinant proteins analyzed by western blot. All data are mean ± s.e.m. *P < 0.05, **P < 0.01, ***P < 0.001, ****P < 0.0001 based on Mann–Whitney test for (**a, b**), one-way ANOVA for (**c–e**) and Two-way ANOVA for (**f, g**). Data are representative of three independent experiments.

panel). These results indicate that BTNL2 in the TME acts on local γδ T cell populations to induce the production of IL-17A by γδ T cells, which in turn promotes tumour immune resistance.

Next, we explored the IL-17A-producing cells in the TME, and found that the tumour-infiltrating IL-17A-producing cells were mainly γδ T cells in CT26 and A20 tumours (Fig. 3a, b). Mouse γδ T cells can be divided into Vγ1, Vγ4, Vγ5, Vγ6, Vγ7 γδ T cell subgroups according to TCR V gene utilization[17], while Vγ4 and Vγ6 γδ T cells have been reported to be the major IL-17A-producing γδ T cells in multiple mouse tumour models[18–21]. Surprisingly, we found that the majority of tumour-infiltrating γδT17 cells were Vγ1 γδ T cells (Fig. 3c, d). Vγ1 γδ T cells were reported as an IL-17A-producing cell in certain circumstances, such as in the lethal pulmonary aspergillosis infections[17,52]. It was reported that IL-17A-producing γδ T cells were CD27 negative, and we indeed found that these subcutaneously infiltrated γδT17 cells did not express CD27, which was consistent with the previous report (Fig. 3c, d)[21,53,54]. Depletion of Vγ1 γδ T cells also had a therapeutic effect on subcutaneous tumours, and following Vγ1 γδ T cell depletion the anti-tumour effect of BTNL2 blockade was comparable to that of controls, which indicates that the anti-tumour effect of BTNL2 blockade is dependent upon Vγ1 γδ T cells (Fig. 3e). Furthermore, we found that Vγ1 γδ T cell depletion decreased IL-17A secretion by CT26 tumour-infiltrating cells, with similar IL-17A secretion seen after treatment with anti-BTNL2 and controls (Fig. 3f). Consistent with our prior findings, these data further confirmed that Vγ1 γδ T cells were the major IL-17A-producing γδ T cells in tumours, and the anti-tumour effect of BTNL2 blockade was dependent on the regulation of Vγ1 γδT17 cells. This finding is also consistent with a previous report that Vγ1 γδ T cells play a tumour-promoting role in the tumourigenesis[55]. As BTNL2 regulates tumourigenesis through Vγ1 γδ T cells, one of the possible mechanisms is that BTNL2 directly binds Vγ1 γδ T cells through Vγ1 γδTCR. To explore this hypothesis, we first examined whether BTNL2 can specifically bind Vγ1 γδ T cells. Interestingly, we did find that BTNL2-FC bound Vγ1 γδ T cells but not Vγ2 and Vγ7 γδ T cells (Supplementary Fig. 4b). Consistent with the finding that BTNL2 inhibited the activation of CD4+ T cells, BTNL2-FC also bound CD4+ T cells (Supplementary Fig. 4b). To examine whether BTNL2 binds Vγ1 γδTCR, we reconstituted the TCR Vγ1.1 and TCR Vδ6.3 in 293 T cells, as it was reported that Vγ1.1 paired with Vδ6.3 in the γδ T cells[56,57]. We did not find that BTNL2 bind TCR Vγ1.1Vδ6.3, and the data indicate that the receptor on γδ T cells is not TCR Vγ1 γδTCR (Supplementary Fig. 4c).

**BTNL2 blockade decreases intra-tumoural accumulation of MDSC populations.** MDSCs were previously shown to be important downstream target cell populations of tumour-promoting γδT17 cells[18,21]. MDSCs can be further subdivided into two subsets: so-called monocytic (CD11b+Ly6G-Ly6C^high) and granulocytic (CD11b+Ly6G+Ly6C^low) MDSCs, which have the same surface markers of monocytes and neutrophils[58,59]. We detected a substantial decrease in tumour infiltration by neutrophils following anti-BTNL2 mAb treatment compared with controls (Fig. 4a). We also detected two distinct CD45+Ly6G− sub-populations that were greatly affected by anti-BTNL2 mAb treatment: a CD45 mid-expression population that was greatly reduced, and a CD45 high-expression population that was significantly increased after anti-BTNL2 mAb treatment. We designated these two populations '1' and '2' for convenience, and found that they were, respectively, positively and negatively associated with tumour size (Fig. 4a). We suspected that the '1' population represented MDSCs, and indeed, found that over 90% of the '1' population highly expressed monocytic MDSCs markers CD11b and Ly6C (Supplementary Fig. 5a). Of the remaining cells, ~1% represented dendritic cells (DCs), and 6% represented macrophages (Supplementary Fig. 5b). Interestingly, almost all of the tumour-infiltrating neutrophils were phenotypically consistent with granulocytic MDSCs, which indicates they are the same cells (Supplementary Fig. 5a). These data indicate that BTNL2 blockade decreases tumour infiltration by both monocytic and granulocytic MDSCs. Consistent with this finding, the CD45+CD11b+Gr-1+MDSC population was in fact reduced following anti-BTNL2 mAb treatment, which held true in CT26 tumour models (Fig. 4b). Given the side- and forward-scatter characteristics of the '2' population, we suspected that these cells were predominantly lymphocytes. Indeed, we found that 85% of the '2' population was composed of CD8+, γδ+ and CD4+ T cells (Fig. 4c). Infiltration of CD45+CD8+ T cells in the TME was greatly increased after anti-BTNL2 mAb treatment compared with anti-control Ab treatment (Fig. 4d). Importantly, the CD8+IFN-γ+ T cell population was significantly increased after BTNL2 blockade (Fig. 4e). BTNL2 inhibition led to decreased MDSC infiltration, but an increased accumulation of CD8+IFN-γ+ T cells, in the TME in the A20 tumour model (Fig. 4f, g). Following depletion of CD8+ T cells in tumour-bearing mice using anti-CD8a neutralizing antibody, the anti-tumour effect of anti-BTNL2 treatment was completely abolished, indicating that CD8+ T cells are the major tumour-cytotoxic cell following BTNL2 blockade (Fig. 4h, i). To further examine the role of MDSCs in the anti-tumour effect of BTNL2 blockade, we depleted MDSCs in tumour-bearing mice using CD11b

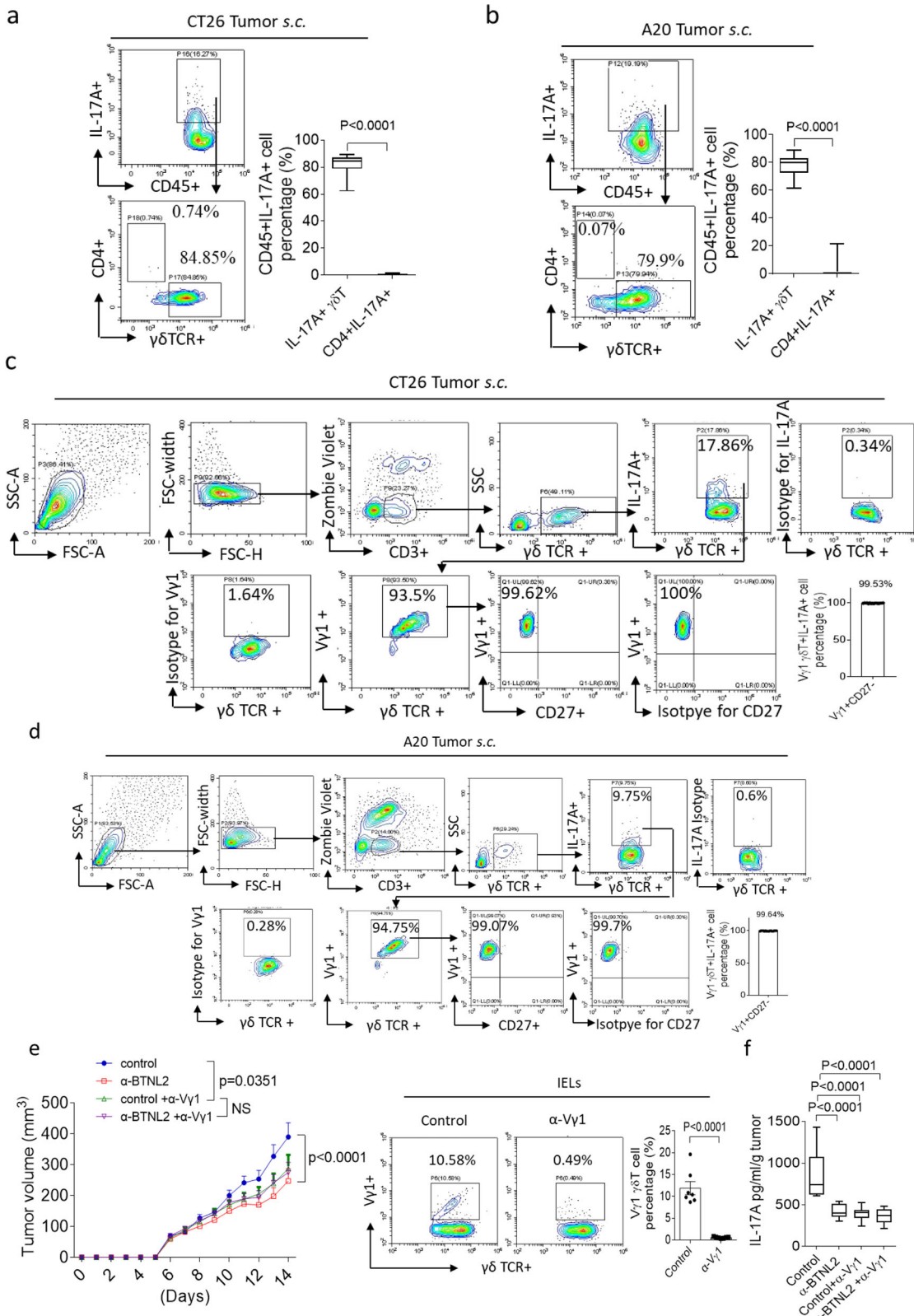

neutralizing antibodies, but did not utilize anti-Gr-1 mAb, as we detected CD8[+] lymphocytes expressing Ly6C in the TME (Supplementary Fig. 5c). Depletion of MDSCs completely abolished the previously observed increase in CD8[+]IFN-$\gamma$[+] T cell tumour infiltration as well as the therapeutic anti-tumour effect of BTNL2 blockade (Fig. 4h–j). Consistent with previous reports, MDSCs

significantly inhibited the proliferation of CD8 + T cells and OVA peptide-stimulated OT-1 CD8 + T cells in the in vitro system (Supplementary Fig. 5d, e). Consistent with this, IL-17A neutralization also led to decreased MDSC and increased CD8[+]IFN-$\gamma$[+] T cells infiltration in the TME (Fig. 4k). Treatment with recombinant IL-17A-Fc protein abolished the BTNL2 blockade-mediated reduction

**Fig. 3 Tumour-infiltrated IL-17A-producing cells were mainly Vγ1 γδ T cells. a, b** Infiltrated cells from subcutaneous tumour of CT26 and A20 were isolated, followed by flow cytometry analysis as indicated ($n = 17$, $P < 0.0001$). **c, d** Infiltrated cells from tumour of CT26 (**c**), A20 (**d**) were isolated, followed by flow cytometry analysis as indicated ($n = 16$ for **c** and $n = 16$ for **d**). **e** Vγ1 γδ T cells were depleted by neutralizing antibody described in the Methods (80 μg/mouse), and CT26 tumour growth kinetics was shown ($n = 13$ for each group, $P = 0.0351$ for control vs control+α-Vγ1, $P < 0.0001$ for control vs α-BTNL2, NS for control+α-Vγ1 vs α-BTNL2 + α-Vγ1). The depletion effect of Vγ1 γδ T cells in the IEL was shown in the right panel ($n = 7$, $P < 0.0001$). **f** CT26 tumours were processed as in **e**, and IL-17A secretion by TME-infiltrated cells was measured by ELISA ($n = 12$, $P < 0.0001$ for control vs α-BTNL2, control+α-Vγ1 and α-BTNL2 + α-Vγ1). All data are mean ± s.e.m. *$P < 0.05$, **$P < 0.01$, ***$P < 0.001$, ****$P < 0.0001$ based on two-sided Mann–Whitney test for (**a**, **b**), Two-way ANOVA for (**e**) and one-way ANOVA for (**f**). Data are representative of three independent experiments.

in MDSC and increase in CD8$^+$IFN-γ$^+$ cell infiltration of the TME (Supplementary Fig. 5f, g). These results further demonstrate that the anti-tumour effect of BTNL2 blockade is mediated at least in part by a reduction in γδT17 cell infiltration in the TME, which as a consequence decreases tumour infiltration by MDSCs, which in turn leads to increased cytotoxic CD8$^+$ T cell infiltration.

**Tumour expression of BTNL2 plays a major role in anti-tumour immune escape.** Next, we explored the physiological role of BTNL2 by examining the phenotype of BTNL2-KO mice. When compared to control mice, BTNL2-KO mice did not show any significant abnormality in survival or weight at 6 months of age (Supplementary Fig. 6a, b). Interestingly, BTNL2 protein was exclusively expressed in the mouse gastrointestinal tract, with particularly robust expression observed in small intestinal epithelial cells (Supplementary Fig. 6c, d). Histologic examination of BTNL2-deficient mice revealed large bowel inflammation, as well as loss of colonic crypt architecture (Supplementary Fig. 6e). Consistent with these findings, the expression of pro-inflammatory cytokines, including *TNFα* and *IL-1β*, was significantly increased in the colonic tissue of BTNL2-KO mice compared with control mice (Supplementary Fig. 6f). Examination of the intestinal epithelial lymphocyte population revealed significantly decreased γδT17 cells in BTNL2-KO mice compared with littermate controls, whereas the IFN-γ-producing γδ T cell population was unchanged (Supplementary Fig. 6g). These data indicate that BTNL2 is a physiologic regulator of γδ T cell differentiation in the gut, and may be utilized by cancer cells to escape anti-tumour immunosurveillance.

We then explored the impact of genetic BTNL2 deficiency on anti-tumour immune responses. Wild-type LLC tumour showed attenuated growth in BTNL2-KO mice compared to that in control mice; however, the difference did not reach statistical significance (Fig. 5a). LLC tumour cells with genetic deletion of BTNL2 (BTNL2-KO LLC) had significantly slower growth in vivo than did wild-type LLC tumours, however there was no difference in tumour cell proliferation between control and BTNL2-KO LLC cells in vitro (Fig. 5b–d). Consistently, BTNL2-KO LLC tumours had significantly fewer numbers of infiltrating γδT17 cells and MDSCs, and conversely higher numbers of infiltrating CD8$^+$IFN-γ$^+$ T cells, as compared to wild-type LLC tumours (Fig. 5e). There was no detectable difference in tumour growth between BTNL2-KO LLC tumours implanted in control mice or BTNL2-KO mice (Fig. 5f). Together, these data indicate that tumour cell-autonomous expression of BTNL2 plays a major role in tumour escape from immunosurveillance in the TME, and we think that the non-tumour cell-expressed BTNL2 plays a less important role in this process. Importantly, prior studies have classified the LLC tumour model as a non-immunogenic "cold" mouse tumour due to the lack of response to anti-PD-1 therapy[60,61]. The significant therapeutic effect against this tumour suggests that antibody-mediated inhibition of BTNL2 may hold promise in combination with anti-PD-1/PD-L1 ICBs, and furthermore, may open the door to immunotherapy for tumours that have previously been considered "cold".

**BTNL2 expression in human cancers correlates with patient's prognosis and γδT17 infiltration.** BTNL2 polymorphisms are associated with susceptibility to lung adenocarcinoma[40,41], and γδT17 cells have been reported to play key roles in human colorectal cancer (CRC) progression through recruitment of MDSCs[23]. We therefore tested the hypothesis that a correlation might exist between BTNL2 expression level and patient outcomes in these two cancer types. Patients with lung adenocarcinoma and colon adenocarcinoma that expressed low levels of BTNL2 had significantly improved survival compared to those expressing high levels of BTNL2 (Fig. 6a, b). We expanded our investigation to include other cancer types, and found that BTNL2 was indeed widely expressed, with expression detected in all of the examined human cancer samples (Supplementary Fig. 7a, b). In unaffected tissue adjacent to cancerous lesions, expression of BTNL2 was significantly decreased compared to that observed in lung adenocarcinoma and colon adenocarcinoma lesions, which is quite similar to the expression pattern of PD-L1 (Supplementary Fig. 7c–e). Importantly, ~38% of lung adenocarcinoma samples had medium-to-high levels of BTNL2 expression, but low levels of PD-L1 expression. This finding suggests that BTNL2 may be a complementary therapeutic target in this subpopulation of patients (Fig. 6c).

In lung adenocarcinoma samples, BTNL2 was mainly expressed by cancer cells (Supplementary Fig. 7f). The abundance of γδT17 cells in freshly isolated lung adenocarcinoma samples was also significantly higher than in para-cancerous samples, and, importantly, protein levels of BTNL2 were much higher in almost all of the examined cancer samples when compared to matched para-cancerous tissue (Fig. 6d–f). Notably, in 7 out of 23 samples (no.1, 3, 7, 9, 13, 15 and 20), tumour lesions had significantly higher expression of BTNL2 than did match para-cancerous tissue, while PD-L1 expression showed no difference. This finding suggests the possibility that BTNL2, more so than PD-L1, may represent a critical mechanism of immune evasion in these tumours (Fig. 6e). Interestingly, in hepatocellular carcinoma, 9 out of 27 samples showed higher tumour expression of BTNL2 compared to matched para-cancerous tissue (no. 1, 14, 15, 18, 19, 22, 24, 25 and 26), while 10 samples (no. 3, 4, 5, 6, 8, 10, 13, 16, 17, 20) showed lower expression of BTNL2 in tumour compared to para-cancerous tissue. In 25 out of 27 hepatocellular carcinoma samples, the pattern of PD-L1 expression (relative expression in cancer vs. para-cancerous tissue) generally reflected the pattern of BTNL2 expression (Fig. 6g). This data suggests that BTNL2 may also play a role in tumour immune escape in hepatocellular carcinoma. Of note, a majority of the hepatocellular carcinoma samples were obtained from patients with chronic hepatitis B virus infection (no. 1–7, 11–27), thus it is possible that BTNL2 expression in the para-cancerous tissue from these samples may not be reflective of expression in uninfected liver tissue.

Next, we analyzed in detail the pattern of BTNL2 expression in lung adenocarcinoma. We found that BTNL2 was robustly and exclusively expressed by tumour cells, and was almost not expressed in the para-cancerous tissue (Fig. 7a, b). Additionally,

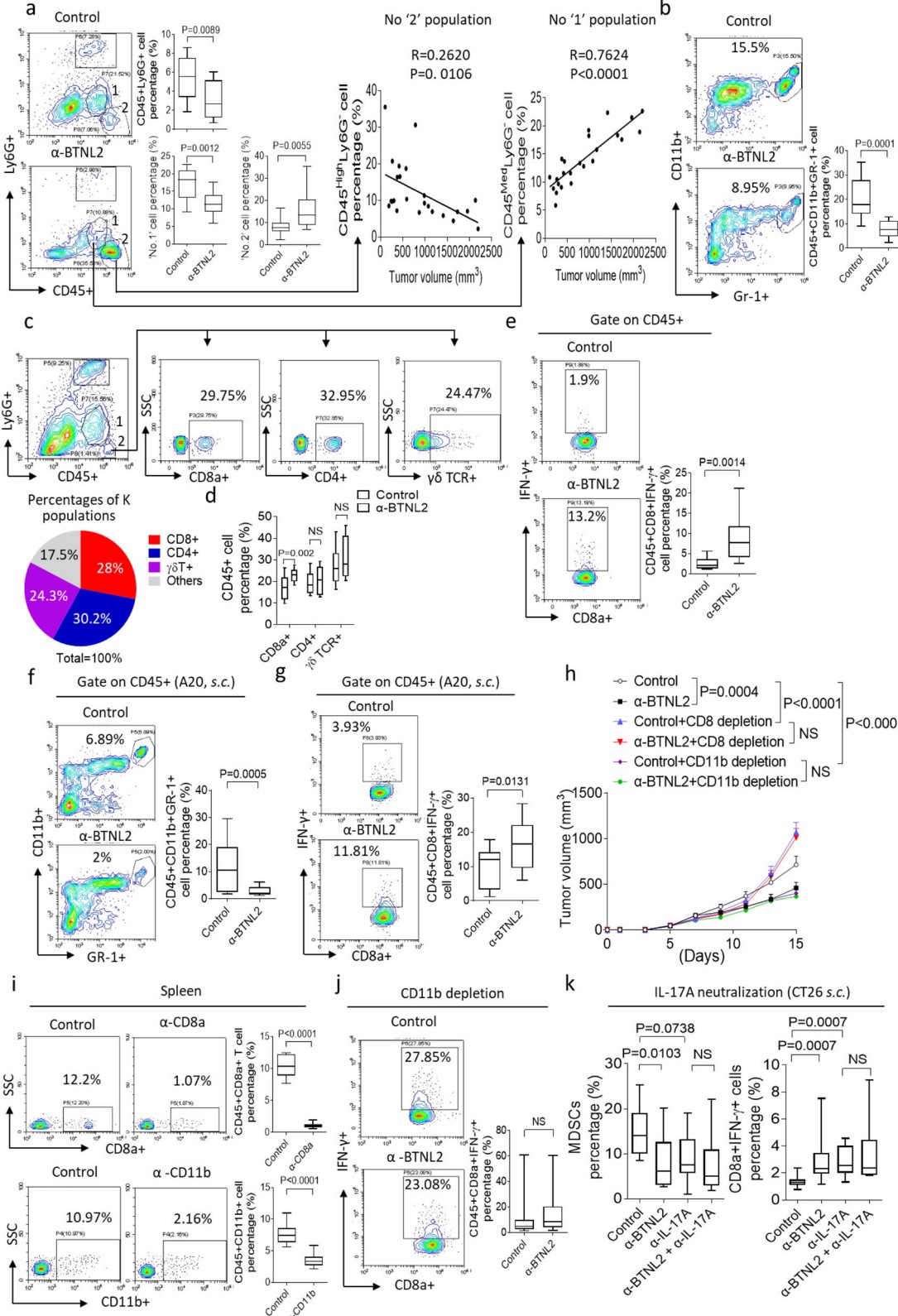

we found that BTNL2 expression reliably mirrored the expression of Napsin A, which is a well-established biomarker for lung adenocarcinoma (Fig. 7a, b). This data suggest that BTNL2 is not only an attractive target candidate for cancer immunotherapy but also may have utility as a biomarker for lung adenocarcinoma.

## Discussion

In addition to the well-described role of CD8[+] T cells in anti-tumour immunity, other cell types have also been shown to play important roles in tumour immunosurveillance, including γδ T cells, and natural killer cells[17,62]. One illustrative study reported

**Fig. 4 Anti-BTNL2 mAb treatment decreases the tumour infiltration of MDSCs. a–e** After isotype control Ab or anti-BTNL2 mAb treatment (200 μg/mouse), infiltrated cells in subcutaneous CT26 tumours were analyzed by flow cytometry as indicated (**a**, $n = 12$, $P = 0.0089$ for CD45$^+$Ly6G$^+$ cell percentage, $P = 0.0012$ for 'No. 1' cell percentage, $P = 0.0055$ for 'No. 2' cell percentage; **b**, $n = 12$, $P = 0.0001$; **d**, $n = 13$, $P = 0.002$ for CD8α$^+$, NS for CD4$^+$ and γδ TCR$^+$; **e**, $n = 13$, $P = 0.0014$). **c** 'No. 2' cell population was further gated by CD4$^+$, CD8$^+$ or γδ TCR$^+$, and the percentages of different cell populations were shown. **f, g** After isotype control Ab or anti-BTNL2 mAb treatment (200 μg/mouse), A20 tumour-infiltrated cell were isolated, followed by flow cytometry analysis ($n = 15$, $P = 0.0005$ for **f**, and $n = 13$, $P = 0.0131$ for **g**). **h** CD8$^+$ T cells or CD11b + cells were depleted by neutralizing antibody as described in the Materials and methods (100 μg/mouse of anti-CD8 or anti-CD11b mAb were used), and CT26 tumour growth kinetics of isotype control Ab or anti-BTNL2 mAb treatment were shown ($n = 16$ for each group, $P = 0.004$ for control vs α-BTNL2, $P < 0.0001$ for control vs control+CD8 depletion and control vs control + CD11b depletion, NS for control+CD8 depletion vs α-BTNL2 + CD8 depletion and control+CD11b depletion vs α-BTNL2 + CD11b depletion). **i** CD8 + T cells or CD11b + cells were depleted by neutralizing antibody (100 μg/mouse), and splenocytes were analyzed by flow cytometry for CD8 + T cells and CD11b + cells ($n = 10$ and 16, $P < 0.0001$). **j** After CD11b cells depletion and control Ab or anti-BTNL2 mAb treatment (100 μg/mouse of anti-CD11b mAb and 200 μg/mouse of anti-BTNL2 mAb were used), infiltrated cells in subcutaneous CT26 tumours were analyzed by flow cytometry as indicated ($n = 13$ for each group, NS for control vs α-BTNL2). **k** After IL-17A neutralization and isotype control Ab or anti-BTNL2 mAb treatment (100 μg/mouse of anti-IL-17A mAb and 200 μg/mouse of anti-BTNL2 mAb were used), infiltrated cells in subcutaneous CT26 tumours were analyzed by flow cytometry as indicated ($n = 15$ for each group, $P = 0.0103$ for control vs α-BTNL2 MDSCs percentage, $P = 0.0738$ for control vs α-IL-17A MDSCs percentage, NS for α-IL-17A vs α-BTNL2 + α-IL-17A MDSCs percentage, $P = 0.0007$ for control vs α-BTNL2 and control vs α-IL-17A CD8α$^+$IFN-γ$^+$ cells percentage, NS for α-IL-17A vs α-BTNL2 + α-IL-17A CD8α$^+$IFN-γ$^+$ cells percentage). All data are mean ± s.e.m. *$P < 0.05$, **$P < 0.01$, ***$P < 0.001$, ****$P < 0.0001$ based on two-way ANOVA for **h**, Dunn's multiple comparisons test for **k**, two-sided Mann–Whitney test for **j** and two-sided unpaired $t$-test for **a–e**, **f**, **g**, **i**. Data are representative of three independent experiments.

that in a cohort of breast cancer patients, tumour-infiltrating γδ T cells were a strong independent predictor of disease recurrence and overall survival. Interestingly, mechanistic studies have shown that human γδ T cells are capable of both tumour-promoting and anti-tumour functions. The anti-tumour role of γδ T cells is thought to be executed by type 1 cytotoxic γδ T cells, which are defined by the expression of IFN-γ. In fact, several clinical trials have employed the adoptive cellular transfer of ex vivo expanded type 1 γδ T cells for the treatment of solid tumours[17]. However, a clear tumour-promoting function of γδT17 cells has also been reported in multiple human solid tumour types, including colorectal cancer, pancreatic cancer and breast cancer[17,19,23,63]. Here, we show that human lung adenocarcinoma samples contain significantly increased numbers of tumour-infiltrating γδT17 cells, when compared to adjacent para-cancerous tissue. This finding suggests that γδT17 cells may also play a tumour-promoting role in lung adenocarcinoma. This also suggests that inhibition of BTNL2-dependent γδT17 cell accumulation holds therapeutic potential in human lung adenocarcinoma, as well as in other γδT17-enriched cancer types (Fig. 7c).

As reported by previous studies, mouse γδ T cells can be divided into Vγ1, Vγ4, Vγ5, Vγ6, Vγ7 γδ T cell subsets, which are defined according to TCR V gene utilization[17]. Several previous reports have suggested that Vγ4 and Vγ6 γδ T cells are the major IL-17A-producing γδ T cell subsets in mouse tumour models[18–21]. However, several groups have reported that Vγ1 γδ T cells are also capable of producing IL-17A, and in one such study, Vγ1 γδ T cells were actually shown to be the dominant source of IL-17A in the lungs of mice during pulmonary aspergillosis infection[17,52]. Vγ1 γδ T cells have also been reported to play critical roles in multiple processes, including tumourigenesis, anti-bacterial host defence and airway hyperresponsiveness[55,64–66]. Interestingly, a previous study utilizing a syngeneic mouse tumour model reported that depletion of Vγ1 γδ T cells had a therapeutic anti-tumour effect, which was mediated by suppression of Vγ4 γδ T cell acitivity[55]. This study is consistent with our finding that the Vγ1 γδ T cell subset has a tumour-promoting effect. In the current study, we found that Vγ1 γδ T cells are the major IL-17A-producing cells in the TME, which differs from prior reports that Vγ4- and Vγ6-restricted cells constitute the dominant γδT17 cell population in tumours. One possible explanation for this discrepancy may be due to differences in mouse tumour models, as we mainly employed heterotopic implantation and intravenous delivery of syngeneic tumour, while the other groups primarily

utilized spontaneous mouse tumour models[18,19,21,67]. Nevertheless, our data from multiple mouse syngeneic tumour models consistently demonstrated a critical role for Vγ1 γδ T cells in tumour progression. Although the precise mechanism by which Vγ1 γδ T cells exert this pro-tumour effect was outside the scope of the present study, we believe that this question merits additional investigation.

The key biomarker for predicting response to anti-PD-1/PD-L1 ICB therapy is the expression of PD-L1 in the TME[68,69]. A recent phase 3 clinical trial enrolled 1274 patients with non-small-cell lung cancer (NSLC) and found that a PD-L1 TPS (tumour proportion score) of 50% or greater predicted a superior response to anti-PD-1 (Pembrolizumab) therapy compared to systemic chemotherapy, while a PD-L1 TPS of 1–49% predicted therapeutic equivalence[70]. Reports have suggested that fewer than half of all NSCLC patients have a TPS of 50% or greater[70], implying that a majority of NSCLC patients fail to realize enhanced clinical benefit from PD-1/PD-L1 blockade over traditional chemotherapy. Our immunohistochemical analysis of lung adenocarcinoma samples from patients revealed that ~60% of tumours had low levels of PD-L1 expression. Of these PD-L1 low-expressors, approximately 68% had medium to high levels of BTNL2 expression, and represented 38.6% of all patients (Fig. 6c). These data suggest that BTNL2 blockade may effectively complement PD-1/PD-L1 inhibition, particularly in the sub-population of patients with low tumour expression of PD-L1 but high expression of BTNL2. Consistent with this, implanted LLC tumours, which are poorly immunogenic "cold" tumours resistant to anti-PD-1 therapy, showed higher expression of BTNL2 compared to PD-L1, and antibody-mediated BTNL2 inhibition yielded a significant therapeutic response (Fig. 1a, Fig. 5; Supplementary Fig. 2c)[71]. This data again supports the notion that BTNL2 is an attractive target with significant potential as an adjuvant to enhance clinical responses to anti-PD-1/PD-L1 ICBs.

Multiple cellular and molecular mechanisms contribute to tumour resistance to ICB therapy, including MDSCs, Treg cells and the expression of immune checkpoint molecules in the TME[3,72,73]. Prior studies have found that patients with cancer who respond poorly to ICB therapy have a high level of circulating MDSCs[25,26]. In patients with prostate cancer and melanoma, MDSCs have also been reported as negative predictors of response to treatment with the anti-CTLA-4 mAb ipilimumab[74,75]. Mirroring these findings in human patients, MDSC numbers have been found to be markedly increased in the TME in multiple mouse tumour models, where they mediate T cell immunosuppression as well as angiogenesis[76]. Therefore, it has been speculated that therapeutic targeting of

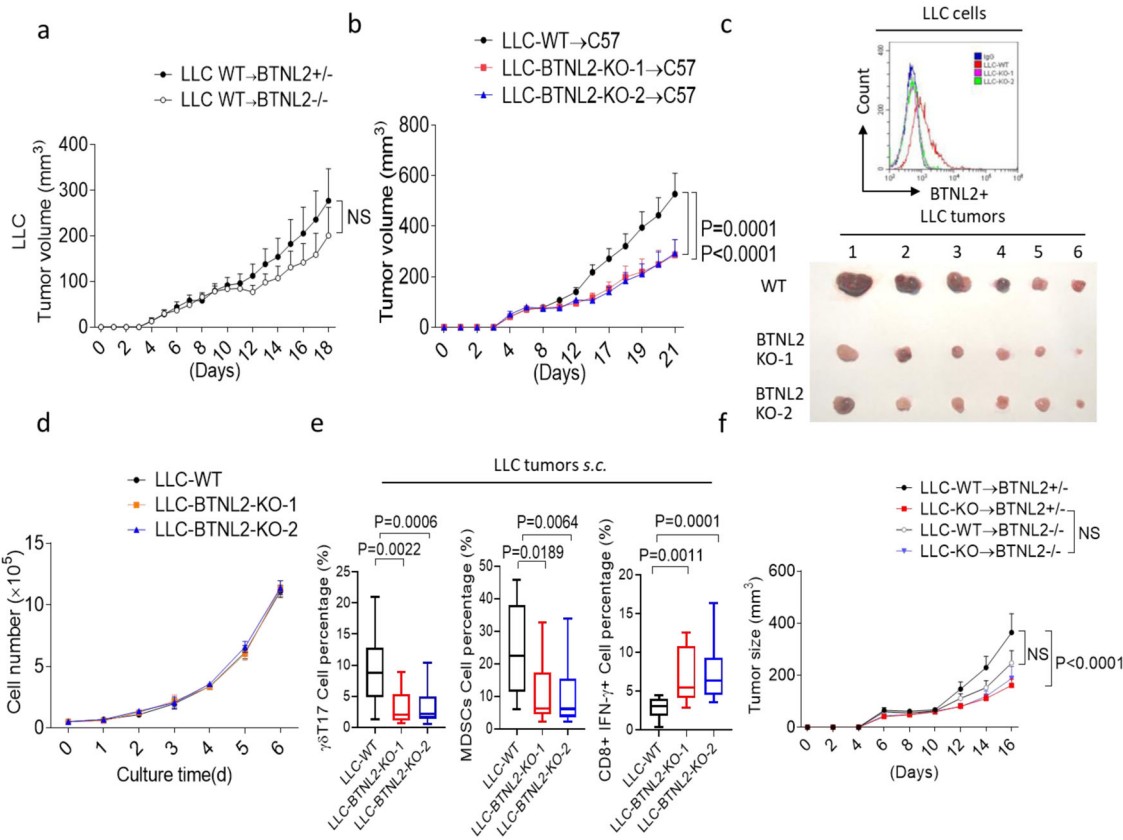

**Fig. 5 Tumour-expressed BTNL2 plays a major role for the anti-tumour immune escaping. a** Littermate control mice or BTNL2-KO mice were implanted WT LLC cells, and tumour growth kinetics of mice was shown (n = 12 for each group). **b** WT LLC cells or two clones of BTNL2-KO LLC cells were implanted subcutaneously in WT mice (3 × 10⁵/mouse), and tumour growth kinetics was shown (n = 14 for each group, P = 0.0001 for LLC-WT vs LLC-BTNL2-KO-1 and P < 0.0001 for LLC-WT vs LLC-BTNL2-KO-2). **c** WT or BTNL2-KO LLC cells were analyzed by anti-BTNL2 mAb-2 for flow cytometry analysis (upper panel). Tumour image from **b** was shown (lower panel). **d** WT LLC cells or BTNL2-KO LLC cells were cultured in vitro for indicated times, and cell proliferation was shown by cell counting. **e** Cells from WT LLC tumours or BTNL2-KO LLC tumours were isolated, followed by the flow cytometry analysis (n = 14 for each group, P = 0.0022 for LLC-WT vs LLC-BTNL2-KO-1 γδT17 cell percentage and P = 0.0006 for LLC-WT vs LLC-BTNL2-KO-2 γδT17 cell percentage, P = 0.0189 for LLC-WT vs LLC-BTNL2-KO-1 MDSCs cell percentage and P = 0.0064 for LLC-WT vs LLC-BTNL2-KO-2 MDSCs cell percentage, P = 0.0011 for LLC-WT vs LLC-BTNL2-KO-1 CD8α⁺IFN-γ⁺ cells percentage and P = 0.0001 for LLC-WT vs LLC-BTNL2-KO-2 CD8α⁺IFN-γ⁺ cells percentage). **f** Littermate control mice or BTNL2-KO mice were implanted with WT or BTNL2-KO LLC cells, and tumour growth kinetics of mice was shown (n = 12 for each group, P < 0.0001 for control mice were implanted with WT vs BTNL2-KO LLC cells, NS for control mice or BTNL2-KO mice were implanted with BTNL2-KO LLC cells and control mice or BTNL2-KO mice were implanted with WT LLC cells). All data are mean ± s.e.m. *P < 0.05, **P < 0.01, ***P < 0.001, ****P < 0.0001 based on two-way ANOVA for **a, b, d, f**, and Dunn's multiple comparisons test for **e**. Data are representative of three independent experiments.

MDSCs might offer significant additive or synergistic benefits when combined with ICB therapy, which is supported by our finding that dual-inhibition of PD-1 and BTNL2 bolsters anti-tumour immune responses (Fig. 1f–h). In the current study, we found that BTNL2 blockade reduced MDSC infiltration of the TME, which we found was primarily mediated by IL-17A production by tumour-infiltrating γδ T cells. However, we cannot exclude the possibility that other cellular populations, for example, M2 macrophages or tumour-associated macrophages (TAM), also act downstream of BTNL2 to modulate anti-tumour immunity, therefore further investigation is needed to clarify this issue. The presence of other immune checkpoint molecules in the TME also undoubtedly contributes to resistance to ICB therapy. As has been reported previously, treatment with anti-PD-1 agents may induce the expression of other checkpoint molecules, such as TIM-3 and LAG3, and dual-blockade of PD-1/TIM-3 or PD-1/LAG3 has been shown to enhance the anti-tumour response[77,78]. Interestingly, we found that anti-PD-1 treatment resulted in upregulation of BTNL2 in the mouse TME (Supplementary Fig. 2d), which mirrors a prior clinical

report that the expression of BTNL2 was upregulated after patients received anti-PD-1 treatment[48].

Here, we have demonstrated that BTNL2 is a potent suppressor of anti-tumour immunity, and acts via a previously unreported role in the regulation of γδT17 cells. However, several questions regarding this promising therapeutic target remain and should be explored in future studies. For example, useful insights will be gained by elucidating the mechanism by which BTNL2 expression is selectively upregulated in the TME, as well as by defining its cognate receptor on γδ T cells. The human and mouse BTNL2 show high protein conservative (64% in terms of amino acid sequence), and both of them contain two extracellular IgC and two IgV domains[36]. The high conservative between human and mouse BTNL2 suggests that human and mouse BTNL2 may function through a similar mechanism, while the function of human BTNL2 needs to be further investigated in the future study. Our findings make clear that BTNL2 is a promising therapeutic target in the ongoing effort to broaden and enhance cancer immunotherapy.

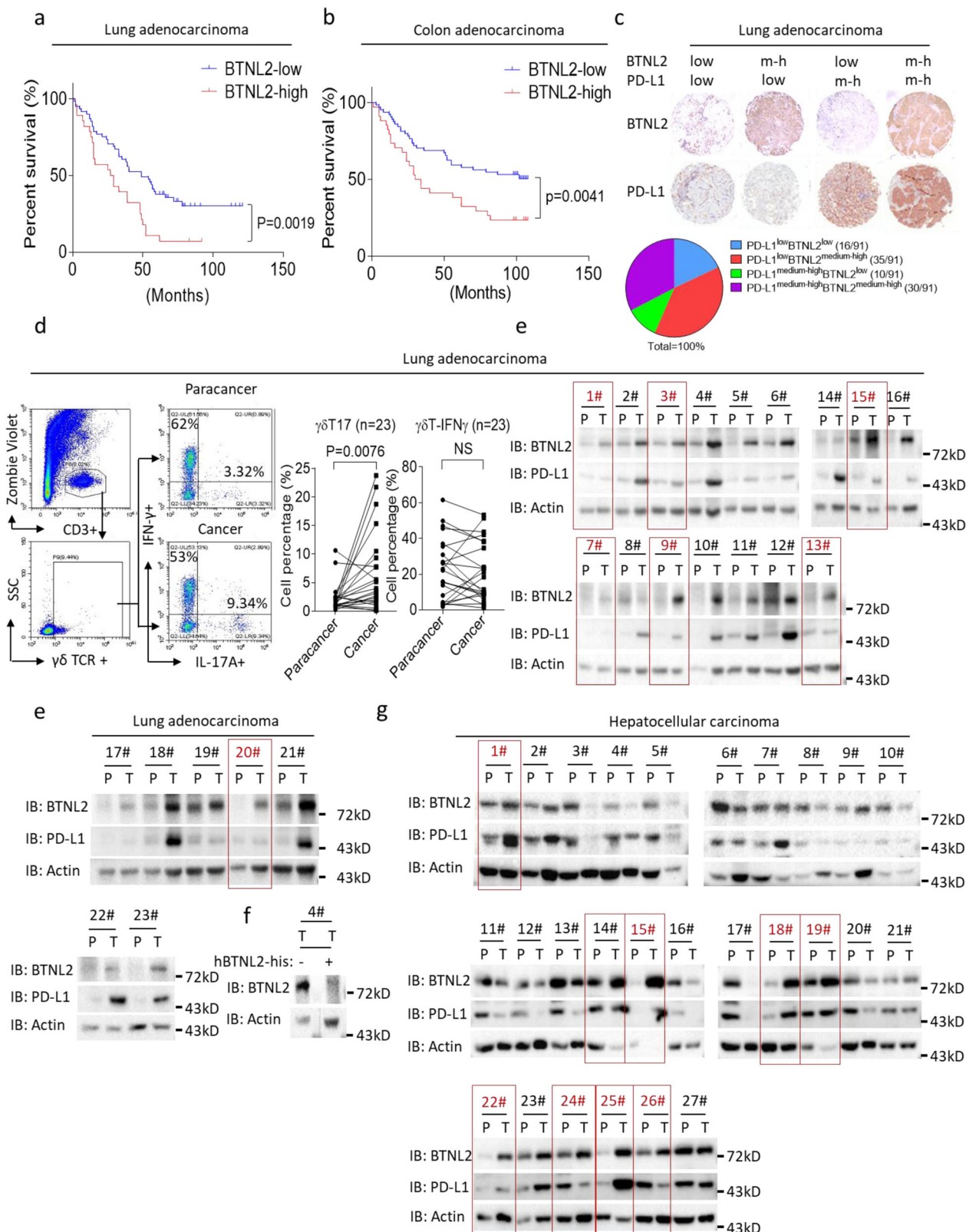

## Methods

**Clinical specimens**. Cancer samples and paired para-cancerous tissue samples were obtained from patients with lung adenocarcinoma who underwent surgical resection at the Department of Thoracic Surgery, Renmin Hospital of Wuhan University. Cancer samples and paired para-cancerous tissue samples were obtained from patients with hepatocellular carcinoma who underwent surgical resection at the department of Hepatic Surgery Center, Tongji Hospital, Tongji

Medical College of Huazhong University of Science and Technology. All samples were anonymously coded in accordance with local ethical guidelines (as stipulated by the Declaration of Helsinki), and written informed consent was obtained. These studies were conducted according to the Declaration of Helsinki and the protocols were approved by the Review Board of the Renmin Hospital of Wuhan University (Approval No: WDRY2019-K063), Tongji Medical College of Huazhong University of Science and Technology (Approval No: S1231).

**Fig. 6 BTNL2 expression in human cancers correlates with patient's prognosis. a, b** Kaplan–Meier estimates of overall survival of lung adenocarcinoma (**a**) and colon adenocarcinoma patients **b** based on the expression level of BTNL2 ($n = 91$ for **a** and $n = 99$ for **b**). Comparison was made of groups with high BTNL2 expression (score ≥ 9) and low BTNL2 expression (score < 9) according to immunohistochemistry scoring system described in the Materials and methods. **c** The percentages of lung adenocarcinoma samples with different expression patterns of BTNL2 and PD-L1 were shown, and score ≥ 5 was considered medium to high expression, and score ≤ 4 was considered low expression. 'm-h' indicates medium to high expression level of BTNL2. **d** Cells were isolated from lung adenocarcinoma samples and para-cancerous samples ($n = 23$), and were stained as indicated for flow cytometry analysis. **e** Lysates from 23 pairs of cancer samples and para-cancerous samples from lung adenocarcinoma patients were analyzed by western blot, and probed for indicated proteins. The samples number with higher expression of BTNL2 but low expression of PD-L1 was marked as red. **f** The BTNL2 Ab which incubated with or without recombinant His-hBTNL2 proteins was probed with the cancer sample from no.4 patient, and western blot of BTNL2 was shown (lower panel). **g** Lysates from 27 pairs of cancer samples and para-cancerous samples from hepatocellular carcinoma patients were analyzed by western blot, and probed for indicated proteins. The samples numbers with higher expression of BTNL2 was marked as red. All data are mean ± s.e.m. *$P < 0.05$, **$P < 0.01$, ***$P < 0.001$, ****$P < 0.0001$ based on Log-rank (Mantel–Cox) test for **a** and **b** and two-sided Mann–Whitney test for **d**. Data in **e–g** are representative of three independent experiments.

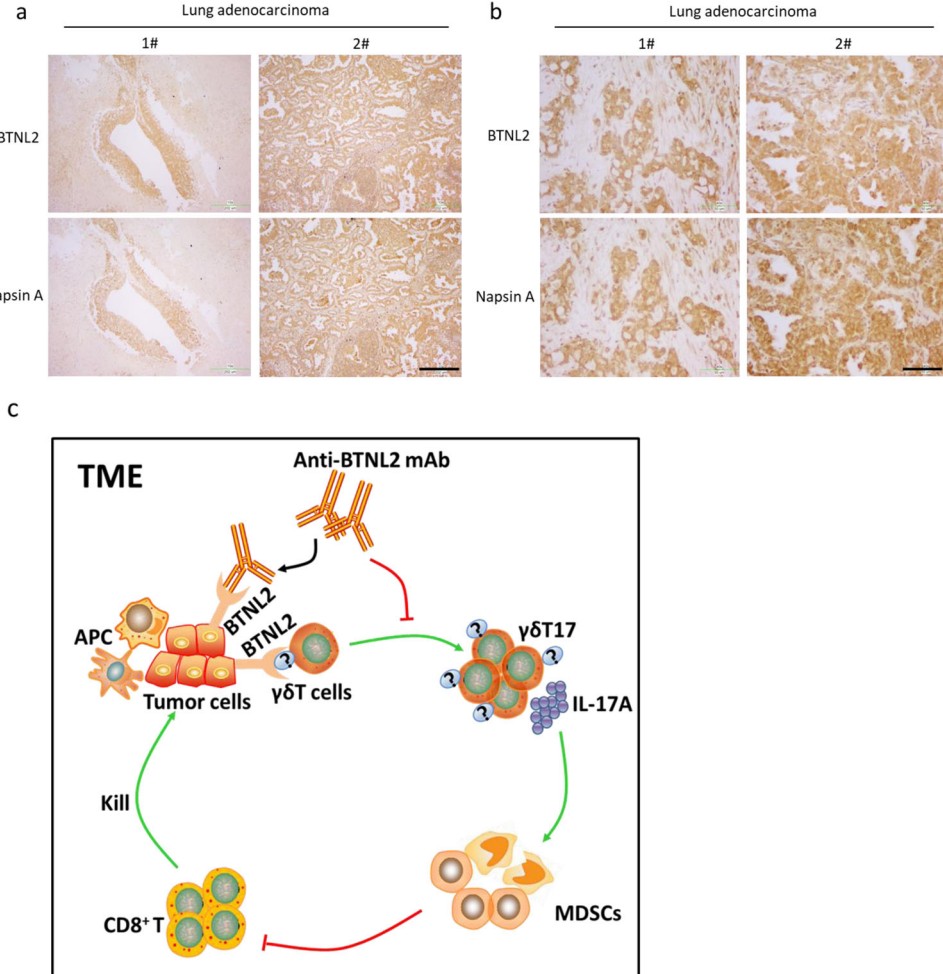

**Fig. 7 BTNL2 was strongly expressed in some of PD-1 inhibitor treatment-resistant lung cancer samples. a, b** Immunohistochemistry staining of BTNL2 and Napsin A expression in lung adenocarcinoma samples was shown (serial section of slices was stained with anti-BTNL2 and anti- Napsin A). Scale bar = 202 μm for **a**, Scale bar = 50 μm for **b**. "P" represents para-cancerous tissue; "C" represents cancer tissue. **c** Sketch Map of BTNL2 blockage in the TME was shown, that tumour cell-expressed BTNL2 promotes γδT17 cells differentiation, which recruits MDSCs into TME to inhibit the cytotoxic function of CD8 + T cells. Blockage of BTNL2 by mAb abolishes the γδT17 cells differentiation and subsequent recruitment of MDSCs, which re-activates CD8 + T cells for the tumour-cytotoxic function. Data in **a**, **b** are representative of three independent experiments.

**Mice**. BTNL2-KO mice were made by Cyagen Biosciences Inc by Crispr-cas9, and the design strategy was shown in Supplementary information, Supplementary Fig. 1b. BTNL2-KO mice were C57BL/6 background. Six- to eight-weeks-old female C57BL/6 and BALB/c mice were obtained from Beijing Vital River Laboratory Animal Technology Co., Ltd. Genotyping primer sequences were shown in the Supplementary Table 1. OT-1 transgenic mice were kindly provided by Prof. Zhengfan Jiang at Peking University and 6–8-weeks-old male OT-1 transgenic mice were used in the experiments. These mice used in our experiments were housed in specific pathogen-free (SPF) condition, the ambient temperature is between 20 and 25 °C, the humidity is

between 40 and 70%, and the environmental light/dark cycle is 12 h light, 12 h dark. All mice were euthanized after experiments. For the euthanasia procedure, put the mice in the euthanasia box and slowly introduce carbon dioxide for 10 min. The mice euthanized by carbon dioxide asphyxiation were checked one by one, followed by cervical dislocation. The experimental protocol was approved by the Institutional Animal Care and Use Committee of Tongji Medical College, Huazhong University of Science & Technology.

To set up the subcutaneous tumour model, $3 \times 10^5$ LLC, $3 \times 10^5$ CT26, $2 \times 10^6$ A20 cancer cells were subcutaneously injected into the right flank of C57BL/6 or

BALB/c (for CT26 or A20 cells) mice on day 0. Rat IgG1 isotype control Ab or anti-BTNL2 mAb were intraperitoneally injected on days 4, 6, 8 and 10 after tumour implantation for 200 μg/200 μl 1×PBS per mouse (tumour size reaches approximate 50–100 mm$^3$). Anti-PD-1 mAb was intraperitoneally injected on days of 4, 6, 8 and 10 of 100 μg/200 μl 1×PBS per mouse. For the depletion of Vγ1 γδ T cells, CD8$^+$ T cells or CD11b$^+$ cells, mice were intraperitoneally injected 80 μg/200 μl per mouse of anti-Vγ1 antibody (polyclonal armenian hamster IgG (BE0091 of BioXcell) as a control antibody), 100 μg/200 μl per mouse of anti-CD8a antibody (rat IgG2b isotype control, anti-keyhole limpet hemocyanin (BE0090 of BioXcell) as a control antibody) or 100 μg/200 μl per mouse of anti-CD11b antibody (BE0090 of BioXcell as a control antibody) at day 1, 3, 6, 10, and 14 after tumour implantation. Isotype control Ab or anti-BTNL2 mAb were intraperitoneally injected at day 4, 6, 8 and 10 after tumour implantation. To neutralize IL-17A in the tumour-bearing mice, mice were intraperitoneal injected 100 μg/200 μl 1×PBS per mouse of anti-IL-17A antibody (murine IgG1 isotype control antibody: BE0083 of BioXcell as a control antibody) at day 0, 4, 6, 9, and 12 after tumour implantation, and anti-BTNL2 mAb were intraperitoneal injected at day 4, 6, 8, 11 and 14 after tumour implantation.

For the experiment of BTNL2 blockage combined with IL-17A-Fc treatment, mice were intraperitoneal injected Fc (recombinant IgG Fc region alone as a control) or IL-17A-Fc recombinant proteins at day 1, 4, 7, 10 and 13 after tumour implantation at dose of 5 μg/mouse. Isotype control Ab or anti-BTNL2 mAb were intraperitoneally injected at day 4, 6, 8 and 10 after tumour implantation.

To set up intravenous mice tumour model, $2 \times 10^5/100$ μl 1×PBS of CT26, MC38 or $2 \times 10^6/100$ μl 1×PBS of A20 cancer cells were injected intravenously, followed by isotype control Ab or anti-BTNL2 mAb treatment at day 1, 3, 6 and 9. For the analysis of tumour-infiltrated cells in the intravenous tumour model, mice were sacrificed and perfused with 1×PBS 18 days after tumour cells intravenously injection. Lungs with tumours were isolated and infiltrated cells were analyzed by flow cytometry.

For the anti-BTNL2 and anti-PD-1 mAb combinational treatment experiment, $4 \times 10^6$ A20 cancer cells were subcutaneously injected into the right flank of BALB/C mice on day 0. Anti-PD-1 (100 μg/200 μl per mouse) or anti-BTNL2 mAb (200 μg/200 μl per mouse) were intraperitoneal injected on days of 4, 6, 8 and 10 after tumour implantation (tumour size reaches ~50–100 mm$^3$). Tumour volume was calculated using the formula L×W$^2$×0.5, where 'L' is the longest dimension and 'W' is the perpendicular dimension. Experimental protocols were approved by the Institutional Animal Care and Use Committee of the Huazhong University of Science & Technology. For the design of the randomization, littermate control mice and BTNL2-KO mice of the same gender were randomly chosen for the in vivo or in vitro experiments.

**Human cancer tissue chips**. Human cancer tissue chips were bought from OUTDO BIOTECH (Lung adenocarcinoma chip cat no. is HlugA180Su05; colon adenocarcinoma chip cat no. is HColA180Su09; liver cancer chip cat no. is HLivH090Su01; breast cancer chip cat no. is HBreD030CS01; esophagus cancer chip cat no. is HEsoS030PG02; gastric cancer chip cat no. is HStmA030PG02; prostate cancer chip cat no. is HPro-Ade045PG-01).

**Reagents**. A rat-anti-mouse BTNL2 monoclonal antibody was made by Atagenix company. In short, five rats were immunized with 6×His tagged mouse BTNL2 27-452aa (extracellular domain of BTNL2) recombinant proteins which purified from 293 F cells. After four time of immunizations, rat splenocytes were fused to mouse myeloma sp2/0 cells, and hybridoma clones were screened by mouse BTNL2-Fc specific ELISA as well as functional experiments showed in Supplementary Fig. 1a. The western blot and flow cytometry were performed by using BTNL2 mAb-2 (Supplementary Fig.1, Supplementary Fig. 2, Supplementary Fig. 6c, d). Isotype control rat IgG1 antibody was bought from R&D Systems (Clone 43414 R, cat no. MAB005R). Anti-human BTNL2 pAb for western blot was bought from Proteintech (1:1000, cat no. 25110-1-AP). Anti-BTNL2 polyclonal antibody for IHC and IF was bought from Sigma (1:100, cat no. HPA039844). The in vivo depletion or neutralizing antibodies of the anti-PD-1 (clone RMP1-14, cat no. BE0146), anti-Vγ1 TCR (clone 2.11, cat no. BE0257), anti-IL-17A (clone 17F3, cat no. BE0173), anti-CD8a (clone 2.43, cat no. BE0061), anti-CD11b (clone M1/70, cat no. BE0007), Polyclonal Armenian hamster IgG (cat no. BE0091), anti-keyhole limpet hemocyanin (clone LTF-2, cat no. BE0090) were bought from BioXcell. Anti-α-Tubulin (1:1000, clone 11H10, cat no. 2125) and Anti-PD-L1 antibodies for IHC were bought from CST (1:200, clone (E1L3N®) XP®, cat no. 13684). Anti-human PD-L1 antibody for western blot was bought from Proteintech (1:2000, clone 2B11D11, cat no. 66248-1-Ig). Anti-Na/K ATPase polyclonal antibody were bought from Proteintech (1:5000, cat no. 14418-1-AP). Anti-β-Actin (1:1000, clone C4, cat no. sc-47778) and anti-HSP90 (1:1000, clone F-8, cat no. sc-13119) antibodies were bought from SANT CRUZ. Anti-Napsin A antibody was bought from Proteintech (1:500, clone 2D12A2, cat no. 60259-2-Ig). Flow antibodies of anti-CD4 (clone RM4-5, cat no. 100510), anti-CD8a (clone 53-6.7, cat no. 100708), anti-IL-17A (clone of TC11-18H10.1, cat no. 506904), anti-IFN-γ (clone XMG1.2, cat no. 505810), anti-IL-17A (clone of TC11-18H10.1, cat no. 506916), anti-IFN-γ (clone XMG1.2, cat no. 505806), anti-IFN-γ (clone XMG1.2, cat no. 505829), anti-γδ TCR (clone UC7-13D5, cat no. 107504), anti-γδ TCR (clone UC7-13D5, cat no. 107512), anti-γδ TCR (clone GL3, cat no. 118116), anti-γδ TCR (clone GL3, cat no.

118124), anti-Vγ1.1TCR (2.11, cat no. 141103), anti-Vγ4 TCR (clone UC3-10A6, cat no. 137703), anti-CD45 (clone of 30-F11, cat no. 103108), anti-CD45 (clone of 30-F11, cat no.103116), anti-GR-1 (clone RB6-8C5, cat no. 108407), anti-Ly6G (clone 1A8, cat no. 127608), anti-F4/80 (clone BM8, cat no. 123115), anti-CD11b (clone M1/70, cat no. 101212), anti-CD11b (clone M1/70, cat no. 101207), anti-Ly6C (clone HK1.4, cat no. 128017), anti-CD11c (clone N418, cat no. 117310), anti-CD16/32 (clone 93, cat no. 101302), anti-Asialo GM1 (clone Poly21460, cat no. 146007), anti-CD3 (clone 17 A2, cat no. 100236), anti-I-A/I-E (clone M5/114.15.2, cat no. 107607) and anti-CD27 (clone LG.3A10, cat no. 124211) were bought from Biolegend. True-Nuclear™ Mouse Treg Flow™ Kit was bought from Biolegend (cat no. 320029). Antibodies of anti-human CD3 (clone HIT3a, cat no. 300306), anti-human γδ TCR (clone B1, cat no. 331222), anti-human IL-17A (clone BL168, cat no. 512306) and anti-human IFN-γ (clone B27, cat no. 502512) were bought from Biolegend. Anti-mouse Vγ7 TCR antibody was kindly provided by Dr. Pablo Pereira at Institut Pasteur in France. Human TruStain FcX™ (Fc Receptor Blocking Solution) was bought from Biolegend (cat no. 422302). Zombie Violet Fixable Viability kit was bought from Biolegend (cat no. 423114). Cell Activation Cocktail (with Brefeldin A) was bought from Biolegend (cat no. 423304). Mouse IL-2 and IL-17A ELISA kits were bought from Biolegend (cat no. 431004 and cat no. 432504). pINFUSE-hIgG2-Fc2 vector was bought from Invivogen (cat no. pfc2-hgin2). PNGase F was bought from NEB (cat no. P0704L). Recombinant murine IL-1β and IL-23 proteins were bought from Sino Biological (cat no. 50101-MNAE and cat no. CT028-M08H). CellTrace™ CFSE Cell Proliferation Kit was bought from Invitrogen (cat no. C34554).

**Cells**. HEK293T, B16F10, LLC and MC38 cells were maintained in DMEM medium plus 10% FBS and 1% Penicillin-Streptomycin. THP-1, 4T1, CT26 and A20 cells were maintained in RPMI-1640 medium plus 10% FBS and 1% Penicillin-Streptomycin. Cells were maintained and amplified in CO$_2$ incubator in a condition of 37 °C, 5% CO$_2$. 293 F cells were maintained in GIBCO FreeStyle 293 Expression Medium (Thermo Fisher, cat no. 12338026), and were maintained and amplified in CO$_2$ shaking incubator in a condition of 37 °C, 5% CO$_2$, 130 rpm.

**Preparation of murine BTNL2-Fc and IL-17A-Fc recombinant protein**. The cDNA sequence coding the murine extracellular portion of BTNL2 protein (aa 27–452) was PCR amplified and subcloned into pINFUSE-hIgG2-Fc2 vector (Invivogen, cat no. pfc2-hgin2) backbone with a human IgG2 Fc tag. The cDNA sequence coding for murine extracellular portion of IL-17A protein (Ala26-Ala158) was PCR amplified and subcloned into pINFUSE-hIgG2-Fc2 vector. The BTNL2-Fc or IL-17A-Fc expression vector was transiently transfected into 293 F cells by using FectoPRO transfection reagent from PolyPlus company according to manufacturer's instruction (cat no. 116-040). Five to six days after transfection, cell supernatants were harvested and purified by affinity chromatography with protein A in accordance with the manufacturer's purification system. SDS-PAGE and Coomassie blue staining were used to analyze the protein preparation, which showed only one major band at the predicted molecular weight.

**Immunohistochemistry**. Formalin-fixed and paraffin-embedded tumour samples were deparaffined, rehydrated, and pre-treated with 3% hydrogen peroxidase in 1×PBS buffer for 20 min. Antigen retrieval in DAKO's antigen retrieval buffer was conducted in a steam cooker for 20 min at 96 °C, followed by slowly cooling down at room temperature. After blocking with DAKO's block buffer, avidin/biotin block, sections were incubated with anti-control or anti-BTNL2 antibody (1:100, Sigma, cat no. HPA039844) for 1 hour at room temperature. After incubation with biotin-conjugated goat anti-mouse/rabbit IgG secondary antibody and streptavidin-HRP, positive signals were visualized by DAB kit (BD pharmingen) and counterstained with Harris hematoxylin (Fisher Scientific).

All of staining was assessed by pathologists blinded to the origination of the samples and subject outcome. The widely accepted German semi-quantitative scoring system in considering the staining intensity and area extent was used. Each sample was assigned a score according to the intensity of the nucleic, cytoplasmic, and/or membrane staining (no staining = 0; weak staining = 1, moderate staining = 2, strong staining = 3) and the extent of stained cells (0% = 0, 1–24% = 1, 25–49% = 2, 50–74% = 3, 75–100% = 4). The final score was determined by multiplying the intensity score with the extent of the score of stained cells, ranging from 0 (the minimum score) to 12 (the maximum score).

For the Supplementary Fig. 7e, control-gRNA- infected cells or BTNL2-gRNA-infected cells (pool of GFP-positive cells) were harvested and washed once with 1×PBS. Cells were fixed by 4% paraformaldehyde at room temperature for 10 min, and were sent to a company for embedding and section. The immunohistochemistry staining protocol was the same as staining for cancer chips samples.

**Immunofluorescence**. Formalin-fixed and paraffin-embedded tumour samples were deparaffined, rehydrated, and pre-treated with 3% hydrogen peroxidase in 1×PBS buffer for 20 min. Antigen retrieval in DAKO's antigen retrieval buffer was conducted in a steam cooker for 20 min at 96 °C, followed by slowly cooling down at room temperature. After blocking with DAKO's block buffer, samples were incubated overnight with primary antibody at 4 °C, followed by three times of washes with ice cold 1×PBS and further stained with fluorophore (Alex488 and Alexa 594) conjugated goat anti-mouse/rabbit IgG secondary antibodies. After

staining, samples were counterstained with DAPI and immersed in a mounting medium before being sealed on a slide with nail polish. Sealed slides were analyzed using Leca TCS-SP microscope with companion software (FV31S-SW).

**Flow cytometry**. Mice tumours were collected, dissociated mechanically, digested with 2 mg/ml Collagenase IV (sigma) and 0.2 mg/ml DNase I (Biofroxx) in serum-free DMEM medium at 37 °C. After 40 min, enzyme activity was neutralized by addition of cold RPMI-1640/10% FBS and tissues were passed through 70 μM cell strainers (Biologix group Limited) and single-cell suspensions in T cell culture medium (RPMI-1640, 10% FBS, 100 IU/ml penicillin, 100 μg/ml streptomycin, 0.5% β-mercaptoethanol) were stimulated with Cell Activation Cocktail (with Brefeldin A) for 4 h (for intracellular staining). After stimulation, cells were incubated with anti-CD16/CD32 (Biolegend) or Human TruStain FcX™ before staining with fluorochrome-conjugated monoclonal antibodies. Cell surface staining was done for 30 min at 4 °C. Intracellular staining was done using a fixation/permeabilization kit (Biolegend). Zombie Violet Fixable Viability kit (1:400; Biolegend) was added to exclude dead cells. Flow cytometry data analysis was performed by using CytExpert. The general flow cytometry gating strategy was shown in Supplementary Fig. 5a.

**ELISA**. CT26 tumours (tumour was cut into small pieces with ~50 mm³ per piece, and 3–4 pieces per 48 well) were cultured in 0.5 ml RPMI medium containing 10% fetal bovine serum (FBS) and antibiotics overnight. Supernatants were collected, centrifuged and analyzed with a murine IL-17A ELISA kit from Biolegend (cat no. 432504). Cytokine concentration was normalized to the weight of tumours in each well.

**In vitro CD4+ T cell assays**. CD4+ T cells from naive C57BL/6 mice were purified from spleens by CD4+ T Cell Isolation Kit from Miltenyi Biotec (cat no. 130-104-454), and the purity of CD4+ T cells was examined by flow cytometry. CD4+ T cells were treated with plate-bound anti-mouse CD3ε (Biolegend, clone 145-2C11, cat no.100340) and anti-mouse CD28 (Biolegend, clone 37.51, cat no.102116) antibodies in the absence or presence of Fc or BTNL2-Fc protein (10 μg/ml of Fc or BTNL2-Fc protein were coated to the plates at room temperature for 2 h after anti-mouse CD3ε and anti-mouse CD28 antibodies coating). IL-2 production was measured 48 h after T cell activation by ELISA. For the screening of BTNL2 blocking mAb, isolated CD4+ T cells were treated with plate-bound anti-mouse CD3ε and/or anti-mouse CD28 antibodies in the presence of Fc or murine BTNL2-Fc recombinant protein, supernatant of hybridoma clones were added into CD4+ T cell culture media when activating T cells.

**γδ T cells isolation and in vitro stimulation**. Spleen and lymph node γδ T cells were sorted by FACS. Isolated cells were cultured with plate-coated Fc, WT-BTNL2-Fc or N4S-BTNL2-Fc recombinant proteins (10 μg/ml of recombinant proteins were coated on the plates at room temperature for 2 h) in the absence or presence of IL-1β (10 ng/ml) and IL-23 (10 ng/ml). After 24 h of stimulation, IL-17A concentrations and *RORC* mRNA were determined by ELISA and real-time PCR. For the experiment of Fig. 2c, splenocytes were cultured in the presence of plate-coated Fc, WT-BTNL2-Fc or N4S-BTNL2-Fc recombinant proteins (10 μg/ml) for 48 h, followed by flow cytometry analysis of γδT17 cells or CD4+ T cells (gated by CD45+).

*In vitro T cell proliferation examination by CFSE*. Spleens from Balb/c naive mice were isolated and passed through 70 μm filters to generate a single-cell suspension. After RBC lysis, CD8 + T cells were sorted by FACS and labelled with 5 μM CFSE (Invitrogen) in pre-warmed 1×PBS for 10 min at 37 °C. The CFSE-labelled CD8 + T cells (2 × 10⁵ cell per well) were co-cultured with purified splenic MDSC (GR-1+, 2 × 10⁵ cell per well) from CT26 bearing mice in a round-bottom 96-well plates pre-coated with 2 μg/ml anti-mouse CD3ε and 5 μg/ml anti-mouse CD28 antibodies. After 48 h, cells were harvested and CFSE signal in the gated CD8 + T cells was measured and analyzed by flow cytometry. Additionally, for OT-1 CD8 + T cell proliferation assays, flow cytometry-sorted MDSC (GR-1+, 1 × 10⁵ cell per well) from LLC bearing mice were co-cultured with 5 μM CFSE-labelled splenocytes (3 × 10⁵ cell per well) from OT-1 mice in the presence of OVA peptide (100 μg/ml) in a round-bottom 96-well plates. Three days later, cells were harvested, and proliferative changes in proliferation were assessed by flow cytometry. The Division Index and %Divided were determined by FlowJo software.

**Isolation of small intestinal intraepithelial lymphocytes (IELs)**. The small intestine was opened longitudinally, freed of Peyer's patches, and washed in serum-free RPMI-1640 medium. Intestinal tissues were cut into 5 mm fragment, transferred to a 50 ml plastic conical centrifuge tube (Jet Biofil) and incubated for 30 min in RPMI-1640 medium supplemented with 2% penicillin/streptomycin, 10% Fetal Bovine Serum, 1 mM DTT and 1 mM EDTA on 37 °C incubator. Then the tissues were passed through a 70 μM strainer to remove tissue pieces and centrifuged on a 40%/80% Percoll density gradient at 2500 rpm for 25 min. IEL were harvested from the 40% to 80% Percoll interface.

**Immunoblot**. Cells were harvested and lysed on ice in lysis buffer containing 0.5% Triton X-100, 20 mM Hepes pH 7.4,150 mM NaCl, 12.5 mM β-glycerophosphate, 1.5 mM MgCl₂, 10 mM NaF, 2 mM dithiothreitol, 1 mM sodium orthovanadate, 2 mM EGTA, 20 mM aprotinin, and 1 mM phenylmethylsulfonyl fluoride for 30 min, followed by centrifuging at 12,000 rpm for 15 min. 2× loading buffer was added to the supernatant, followed by boiling for 10 mins.

**Membrane and cytoplasm fractionation**. Intestinal epithelial cells were rinsed with cold 1×PBS for three times and washed once in hypotonic buffer (10 mM KCl, 1.5 mM MgCl2, 10 mM Tris-HCl pH 7.5) supplemented with a protease inhibitor, incubated on ice in hypotonic buffer for 15 min and then pipetted up and down for 5–10 times. The lysates were centrifuged at 4 °C for 5 min at 2500 × g to remove nuclei and cellular debris. Supernatants were centrifuged at 100,000 × g for 60 min at 4 °C to separate cytosolic extracts (S100) and pellets (P100). The pellets (P100) were resuspended in lysis buffer volumes equal to those of the supernatants (S100), stored with the addition of 5 × Loading Buffer, and analyzed by western blot.

**Real-time PCR**. Total RNA was extracted from spinal cord with TRIzol (Invitrogen) according to the manufacturer's instructions. 1 μg total RNA for each sample was reverse transcribed using the SuperScript® II Reverse Transcriptase from Thermo Fisher Scientific. The resulting complementary DNA was analyzed by real-time PCR using SYBR Green Real-Time PCR Master Mix. All gene expression results were expressed as arbitrary units relative to expression *Actb*. Real-time PCR primers were listed in the Supplementary Table 1.

*Lentivirus-medicated gene knockout/knockdown in LLC or THP-1 cells*. pLentiCRISPR-GFP vector was used for CRISPR/Cas9-mediated gene knockout in LLC and THP-1 cell lines. Briefly, lentivirus vector expressing gRNA was transfected together with package vectors into HEK293T (ATCC) package cells. Forty-eight and seventy-two hours after transfection, virus supernatants were harvested and filtrated with 0.2 μm filter. Target cells were infected twice and sorted by flow-cytometry-medicated cell sorting. Single cell was sorted into 96-well plate by flow cytometry for single clone isolation (LLC cell lines). Isolated single clones were verified by DNA sequencing and flow cytometry analysis (Fig. 5c). In some cases, pool of GFP-sorted cells was used in the experiments (THP-1 cell lines). The gRNA sequences for making LLC or THP-1 KO cell lines are listed in Supplementary Table 1.

**Statistics**. Normality distribution of the data was determined by using D'Agostino & Pearson test. Statistical significance between two groups was determined by unpaired two-tailed *t*-test or Mann–Whitney test; multiple-group comparisons were performed using one-way ANOVA or Dunn's multiple comparisons test; Statistical significance for the tumour growth kinetics curve were analyzed with two-way ANOVA. Survival rate was analyzed by Log-rank (Mantel-Cox) Test. $P < 0.05$ was considered to be significant, and $*P < 0.05$, $**P < 0.01$, $***P < 0.001$, $****P < 0.0001$. Results are shown as mean and the error bar represents the standard error of mean (S.E.M) technical or biological replicates as indicated in the figure legend. Graphpad Prism Version 8 was used to analyze the data.

**Reporting summary**. Further information on research design is available in the Nature Research Reporting Summary linked to this article.

## Data availability
The source data underlying Figs. 1–6, Supplementary Figs. 1–7 are provided as a Source Data file. Source data are provided with this paper.

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

## Acknowledgements

We thank Shuyan Liang and Zhixin Qiu in the Wuhan Biobank Co., Ltd for their kind help in the flow cytometry analysis. We thank Dr. Zhengfan Jiang at Peking University for the providing of OT-1 transgenic mice. This investigation was supported by the grant from the Original Exploration Program of National Natural Science Foundation of China (82150102, to C.H.W.); the National Key Research and Development Program of China (2020YFA0710700, to C.H.W.); the Key Research and Development Program of Sichuan province (22ZDYF3738, to C.H.W.); the Fundamental Research Funds for the Central Universities, HUST (2021GCRC031, to C.H.W.); the Junior Thousand Talents Program of China (to C.H.W.); the National Natural Science Foundation of China (81871280, to C.H.W.).

## Author contributions

Y.Y.D. and Q.W.P. performed the experiments with the assistance from T.P., W.W.S., H.P.W., X.J.M., R.R.H., Z.H.C., X.F., Z.Q.L., T.X.Z. and S.Y.L; D.C., L.S. and W.Y.J provided the lung adenocarcinoma samples and helped to process the samples; Z.G.Z. provided the hepatocellular carcinoma samples; N.G. helped to perform the statistical analysis; W.J.H., B.N.M. and C.J.Z. participated the discussion and provided reagents; Y.Y.D., Q.W.P. and C.H.W. analyzed the data; C.H.W. wrote the manuscript and supervised the project.

## Competing interests

The authors declare no competing interests.
