## [Peer Review File · Nature Communications]

Cancer cell-expressed BTNL2 facilitates tumour immune escape via engagement with IL-17A-producing $\gamma\delta$ T cellsREVIEWER COMMENTS

Reviewer #1 (Remarks to the Author):

Du et al. explore the clinical utility of anti-BTNL2 mAb in mice, and in summary (i) identify tumor-associated expression of BTNL2 in mice, and clinical correlation of high BTNL2 on some human carcinomas with poor prognosis; (ii) anti-BTNL2 mAb have anti-tumor efficacy in a number of settings, (iii) identify a key role for Vg1+ IL-17 producing gamma-delta T cells in efficacy of these mAb; and (iv) demonstrate that they reduce myeloid suppressor cells in the tumor microenvironment.

The authors have collated a large body of work and it was enjoyable to read. The strengths of this study are that it is comprehensive, and includes both mouse and human data that appears consistent with the conclusion that BTNL2 is an important regulator of anti-tumor immunity. It also utilizes a range of different mouse tumor models.

I have three significant points to raise, plus some minor comments:

1. The link with V gamma 1+ gamma-delta T cell function via IL-17 is tantalizing, especially since other subsets of gd T cells can directly interact with BTNLs, eg Btn11 with Vg7+ cells. Despite the indication in the discussion that this falls outside the scope of the manuscript, it would greatly benefit the study if the authors can try to explore the question of whether BTNL2 directly binds Vg1+ gdTCR, since it would tie everything together. They seemingly have all of the tools including BTNL2-Fc and established assays of identifying Vg1+ cells. Do Vg1+ gd T cells selectively bind BTNL2-Fc compared to other gd T cell subsets and ab T cells?

2. In some parts, the manuscript would be enhanced by improved technical robustness. For example: (a) with respect to statistics, parametric statistical tests are used throughout. Do the data pass a normal distribution assessment? If not, non-parametric tests that do not assume normal distribution would be more appropriate.

(b) In figs 1-5, the reproducibility statements indicate "data are representative of 3 independent experiments". Is this the case for every panel, in figures 1-5? Ideally, the data from all 3 experiments should be shown, since in many instances the sample sizes depicted within groups is quite low (n=5 or 6), and additional data points would add a stronger degree of confidence. If there is inter-experimental variability, the graphs could show symbols with different colors for the separate experiments, or alternatively the data could be normalized in some way.

(c) Provide detail of what statistical test is used in fig 4a, b d, e and f, g, l, j; also 4h; 4k, and if a parametric test is used, justify it.

3. A separate supplementary figure that shows reactivity of anti-BTNL2 mAb to BTNL2 by ELISA (compared to a control protein) and also reactivity of these same mAb to BTNL2+ cells. This would provide important information about the specificity and selectivity of each mAb. Likewise, a separate supp fig panel depicting the purity and homogeneity of BTNL2-Fc would be useful.

Minor comments/suggestions:

Page 3 line 23. Perforin and Granzyme should be lowercase

Page 3 line 35/36; page 6 line 9 & 10. Should be butyrophilin-like instead of butyrophilins-like

A previous study reported that a BTNL2-Fc 2 fusion protein directly inhibited CD4+ T cell activation in vitro, although the cognate receptor for 3 BTNL2 on T cells has remained unknown^{36, 37}.

- Two primary studies are cited so the sentence should read "Previous studies have..."

Supp Fig 1a

- Define what "Fc" is in the figure. I suspect it is referring to recombinant Fc region alone but it is

unclear.

-Why would some anti-BTNL2 mAb abrogate IL-2 production from CD4 T cells?

Supp Fig 1c. Why does the mBTNL2-Fc migrate at a higher apparent MW than the endogenous BTNL2, which is presumably larger in size due to the transmembrane and intracellular domains?

Supp Fig 1d. The MWM are truncated on the RHS

Page 5 line 8. Should read deletion rather than depletion

Supp Fig 2a. What is the significance of short exposure and long exposure? There is no apparent mention of this in the main text.

Supp fig 2f. What does "N4S" refer to? There is mention of four separate Asn-Ser mutations, but is this the quadruple mutant?

Page 6 line 7 regarding supp fig 2g. "...however a small minority of CD45+ leukocytes did also express BTNL2"

- 49% of CD45+ cells is not a small minority, it is almost half.

Page 6 line 16 ". Notably, there was no significant difference in the number of 16 tumor-infiltrating IL-17A-or IFN- γ -producing CD4+T cells, Tregcells, or NK cells"

- should make reference to the model, eg following CT26 challenge s.c.

Fig 2c, it should be made clear in the figure legend if the cells have been restimulated in order to induce IL-17 expression.

Page 6 line 20. "Notably, the glycosylation site mutant 20 N4S-BTNL2-Fc promoted significantly greater numbers of $\gamma\delta$ T17 cells and production of IL-17A 21 than did WT BTNL2, either with or without co-stimulation with IL-1 β and IL-23"

- the difference is marginal. Could it be influenced by e.g. higher affinity of deglycosylated protein for the plastic, or slight difference in quantification of the BTNL2-Fc, or slightly higher purity of deglycosylated protein?

What is the tumour model used in fig 2f?

Page 9 line 9 "Importantly, 8 prior studies have classified the LLC tumor model as a non-immunogenic "cold" mouse tumor due to 9 the lack of response to anti-PD-1 therapy."

-missing a supporting reference

Discussion: "The importance of the $\gamma\delta$ T cell population was further highlighted by a large 6 bioinformatics-based analysis that included thousands of patients and multiple tumor types, which 7 found that the best correlate for overall survival was the presence of tumor-infiltrating $\gamma\delta$ T cells"

- the accuracy of the gd gene signature has been challenged by Tosolini et al (doi: 10.1080/2162402X.2017.1284723) so the authors may wish to qualify this statement.

Methods

Antibodies used for in vivo depletion, clone names are missing and their source.

Page 15 line 13. "For the design of the randomization, littermate control mice and BTNL2-KO mice of the same 13 gender were randomly chosen for the in vivo or in vitro experiments."

- what is 'randomly chosen' referring to? Were the mice age and sex-matched in each assay? Was this information recorded and can the authors confirm the sex and age of the mice for each experiment?

Reviewer #2 (Remarks to the Author):

Nature Communications manuscript NCOMMS-21-14489

Title: BTNL2 is a suppressor of antitumor immunity by regulating IL-17A-producing $\gamma\delta$ T cells

The data in this manuscript show that butyrophilin like 2 (BTNL2) plays a role in tumor progression by stimulating IL-17A expression in $\gamma\delta$ T cells that encourage neutrophils to suppress CD8 T cells. The authors show that inhibition of BTNL2 through a newly described antibody or genetic deletion of BTNL2 in cancer cells slows the growth of transplanted cancer cell lines in mice. Using in vitro assays, the authors show that BTNL2 can directly stimulate IL-17A production in $\gamma\delta$ T cells. In mice, decreased IL-17A by inhibition of BTNL2 reduces neutrophils and increases cytotoxic T cells, whose contribution is shown by depleting antibodies. Association of BTNL2 expression in human tumor tissue microarrays and cancer patient survival is shown for lung and colon cancer.

Butyrophilins are an understudied family of proteins, so this manuscript provides novel information on its function in cancer immunology. Overall, this is a nice study with good mechanistic insight. The authors use both loss-of-function and gain-of-function assays to support their conclusions. The experiments include proper controls (except in one case, see below) and statistics are appropriate. I only have a few comments to improve the quality of the manuscript:

1. The format of the panel figures needs better organization. It is often difficult to find the panels referenced in the text. Sometimes they go down, sometimes they go across. I realize that there is a lot of data, but the panels need to be easy to follow.
2. The MDSC acronym is not defined in Introduction.
3. Supp Figure 2f is confusing. According to Supp Figure 1d, the 95 kDa band is non-specific, but point mutations in BTNL2 change the molecular weight of the 95 kDa band. How do the authors reconcile this discrepancy? The authors should show the bands between 55 and 72 kDa.
4. In regard to Figure 2, I do not agree with the authors' conclusion of skewed differentiation towards gdT17. The data show that IL-17A is regulated by BTNL2 – they do not show cell differentiation.
5. The sentence describing the data in Figure 2f is incorrect. There is no additive or synergistic effect of anti-IL17A and anti-BTNL2.
6. There is no explanation of the blot in Figure 2g. Please provide description in text.
7. It is surprising to see Vg1 cells producing the majority of IL-17A in the transplant models, as Vg4 and Vg6 cells usually produce the most IL-17A. Because this is an unusual observation, the authors need to show the full gating strategy for these data with controls for Vg1 and CD27 staining. I wonder whether BTNL2 may specifically bind the Vg1 T cell receptor, so the authors may consider this for future work.
8. The authors state that neutrophils and MDSCs are different populations (page 7). This is incorrect. There is no way to distinguish a neutrophil from an MDSC by cell surface molecules, because these are in fact the same cells. The authors should be aware that use of the MDSC terminology is being discouraged by experts (see Hegde, Leader & Merad, Immunity 2021) because it is ambiguous. I strongly suggest that the authors remove any mention of MDSC and refer to the CD11b+Ly6G+ population as neutrophils.

Point-by-point response

We really thank both reviewers' comments and suggestions, which helped us to further improve the quality of this manuscript.

Reviewer #1 (Remarks to the Author):

Du et al. explore the clinical utility of anti-BTNL2 mAb in mice, and in summary (i) identify tumor-associated expression of BTNL2 in mice, and clinical correlation of high BTNL2 on some human carcinomas with poor prognosis; (ii) anti-BTNL2 mAb have anti-tumor efficacy in a number of settings, (iii) identify a key role for Vg1+ IL-17 producing gamma-delta T cells in efficacy of these mAb; and (iv) demonstrate that they reduce myeloid suppressor cells in the tumor microenvironment.

The authors have collated a large body of work and it was enjoyable to read. The strengths of this study are that it is comprehensive, and includes both mouse and human data that appears consistent with the conclusion that BTNL2 is an important regulator of anti-tumor immunity. It also utilizes a range of different mouse tumor models.

We thank the reviewer's comments!

I have three significant points to raise, plus some minor comments:

1. The link with V gamma 1+ gamma-delta T cell function via IL-17 is tantalizing, especially since other subsets of gd T cells can directly interact with BTNLs, eg Btl1 with Vg7+ cells. Despite the indication in the discussion that this falls outside the scope of the manuscript, it would greatly benefit the study if the authors can try to explore the question of whether BTNL2 directly binds Vg1+ gdTCR, since it would tie everything together.

We thank the reviewer's comments and suggestions. We examined whether BTNL2 directly bind V γ 1 $\gamma\delta$ TCR by reconstituting V γ 1.1TCR and V δ 6.3TCR in 293T cells (it was reported that V γ 1.1 paired with V δ 6.3 in the $\gamma\delta$ T cells, PMID: 19416854 and 31356163). We didn't find that BTNL2-Fc directly bind V γ 1.1V δ 6.3TCR, and please find the data in the **Sup Figure 4c** of the revised manuscript.

As BTNL2 also binds CD4⁺ T cells and inhibits the activation of CD4⁺ T cells (**Sup Figure 4b**), the receptor of BTNL2 may not be V γ 1 $\gamma\delta$ TCR, as CD4⁺ T cells don't express this molecule.

They seemingly have all of the tools including BTNL2-Fc and established assays of identifying V γ 1+ cells. Do V γ 1+ $\gamma\delta$ T cells selectively bind BTNL2-Fc compared to other $\gamma\delta$ T cell subsets and ab T cells?

We thank the reviewer's suggestion, and we examined the binding between BTNL2-Fc and V γ 1+, V γ 2+, V γ 7+ $\gamma\delta$ T cells, CD4+ T cells. We found that BTNL2 specifically binds V γ 1+ $\gamma\delta$ T cells and CD4+ T cells, but not binds V γ 2+ and V γ 7+ $\gamma\delta$ T cells (**Sup Figure 4b**).

2. In some parts, the manuscript would be enhanced by improved technical robustness.

For example:

(a) with respect to statistics, parametric statistical tests are used throughout. Do the data pass a normal distribution assessment? If not, non-parametric tests that do not assume normal distribution would be more appropriate.

We thank the reviewer's comments and suggestions, and we examine the normality distribution of the data throughout the manuscript by using D'Agostino & Pearson test, and we found that the data of Figure 2a, 2b, 3a, 3b, 4j, 4k, 5e, 6d didn't pass the normality distribution assessment, and all the other data passed the normality distribution assessment. We re-analyzed these data by using the Dunn's multiple comparisons test (Figure 4k and 5e) and the Mann-Whitney test (Figure 2a, 2b, 3a, 3b, 4j and 6d). We revised the figure legends of Figure 2a, 2b, 3a, 3b, 4j, 4k, 5e, 6d in the revised manuscript.

(b) In figs 1-5, the reproducibility statements indicate “data are representative of 3 independent experiments”. Is this the case for every panel, in figures 1-5? Ideally, the data from all 3 experiments should be shown, since in many instances the sample sizes depicted within groups is quite low (n=5 or 6), and additional data points would add a stronger degree of confidence. If there is inter-experimental variability, the graphs could show symbols with different colors for the separate experiments, or alternatively the data could be normalized in some way.

We thank the reviewer’s suggestions. We now present pooled data from three independent experiments (Figure 1f and 1h were pooled from two independent experiments), and please find the revised data in **Figure 1-5** of the revised manuscript.

(c) Provide detail of what statistical test is used in fig 4a, b d, e and f, g, I, j; also 4h; 4k, and if a parametric test is used, justify it.

We thank the reviewer’s suggestions, and we provided detailed statistical methods for the figure 4a, b, d, e, f, g, i, j, h and k in the figure legends of the revised manuscript.

We used D'Agostino & Pearson test to determine the normality distribution of the data, and we found that all the data except Figure 4j and 4k passed the normality distribution test. We re-analyzed Figure 4j by using Mann-Whitney test and Figure 4k by Dunn's multiple comparisons test, and updated the statistical method in the figure legend of the revised manuscript.

3. A separate supplementary figure that shows reactivity of anti-BTNL2 mAb to BTNL2 by ELISA (compared to a control protein) and also reactivity of these same mAb to BTNL2+ cells. This would provide important information about the specificity and selectivity of each mAb.

We thank the reviewer’s suggestion. We performed western blot and blocking experiment to assess these five mAbs.

We examined whether these five mAbs can do be used for western blot or not, and we

found that all the mAbs except mAb-1 can be used for the western blot, and please find the data below. For the blocking experiment, please find the data in the **Sup Figure 1a**.

We used supernatant of hybridoma clones provided by the company to perform the western blot and blocking experiments.

We also performed the flow cytometry by using purified anti-BTNL2 mAb-2 antibodies (**Sup Figure 1e**). Since the company didn't give us the hybridomas of the mAb-1, 3, 4, 5, we didn't perform flow cytometry by using these clones (it needs to purify the antibodies from the supernatants of the hybridomas to do the flow cytometry experiment).

Likewise, a separate supp fig panel depicting the purity and homogeneity of BTNL2-Fc would be useful.

We thank the reviewer's suggestion. We performed the Coomassie Blue Staining for the recombinant proteins of WT-BTNL2-Fc and N4S-BTNL2-Fc, and please find the data in the **Sup Figure 4a** of the revised manuscript. The purity of WT-BTNL2-Fc and N4S-BTNL2-Fc is quite high, and there aren't obvious nonspecific bands on the gel.

Minor comments/suggestions:

Page 3 line 23. Perforin and Granzyme should be lowercase

We are sorry for the mistakes, and we corrected them in the revised manuscript.

Page 3 line 35/36; page 6 line 9 & 10. Should be butyrophilin-like instead of butyrophilins-like

We thank the reviewer's reminder, and we corrected it in the revised manuscript.

A previous study reported that a BTNL2-Fc fusion protein directly inhibited CD4⁺ T cell activation in vitro, although the cognate receptor for BTNL2 on T cells has remained unknown^{36, 37}.

- Two primary studies are cited so the sentence should read "Previous studies have..."

We thank the reviewer's reminder, and we corrected it in the revised manuscript.

Supp Fig 1a

- Define what "Fc" is in the figure. I suspect it is referring to recombinant Fc region alone but it is unclear.

We are sorry not describing it clearly, and we revised it as "Fc represents recombinant IgG Fc region alone as a control" in the figure legend of Sup Figure 1 of the revised manuscript.

-Why would some anti-BTNL2 mAb abrogate IL-2 production from CD4 T cells?

We thank the reviewer's question. In this experiment, the abrogation of IL-2 production by CD4⁺ T cells was due to the application of BTNL2-Fc recombinant proteins (BTNL2-fc inhibits the activation of CD4⁺ T cells), and the anti-BTNL2 mAb-2 completely reverse the BTNL2-Fc-mediated CD4⁺ T cell inhibition, while some of the clones (mAb-1 and mAb-5) didn't show any impact on the BTNL2-Fc-mediated inhibition of CD4⁺ T cells, and some of the other clones partially reverse the inhibition by BTNL2-Fc on CD4⁺ T cells (mAb-3 and mAb-4).

Supp Fig 1c. Why does the mBTNL2-Fc migrate at a higher apparent MW than the endogenous BTNL2, which is presumably larger in size due to the transmembrane and intracellular domains?

We thank the reviewer's question. For the data in the Sup Figure 1c, murine extracellular portion of BTNL2 protein (aa 27–452) was constructed into pINFUSE-hIgG2-Fc2 vector, which contains human IgG2 Fc tag. The IgG2 Fc contains 223 amino acids, so the BTNL2-Fc has a higher molecular weight than the endogenous BTNL2 on the protein gel. Please note that BTNL2 is a glycosylated protein, and its actual size on the protein gel is higher than that of predicted size due to glycosylation.

Supp Fig 1d. The MWM are truncated on the RHS

We thank the reviewer's comments, and we included the uncut markers in the Supp Figure 1d of the revised manuscript.

Page 5 line 8. Should read deletion rather than depletion

We thank the reviewer's comment, and we revised the manuscript as suggested by the reviewer.

Supp Fig 2a. What is the significance of short exposure and long exposure? There is no apparent mention of this in the main text.

We thank the reviewer's question. Since there isn't much significance to show both short exposure and long exposure, we deleted the data of the short exposure in the revised manuscript.

Supp fig 2f. What does "N4S" refer to? There is mention of four separate Asn-Ser mutations, but is this the quadruple mutant?

We are sorry for the unclear description, and the "N4S" represents the quadruple mutant of BTNL2, in which all the four asparagine were replaced by serine. We

clarified it in the figure legend of the revised manuscript.

Page 6 line 7 regarding supp fig 2g. "...however a small minority of CD45+ leukocytes did also express BTNL2"

- 49% of CD45+ cells is not a small minority, it is almost half.

We thank the reviewer's comments, and we revised the sentence as "however 48.47% of CD45⁺ leukocytes did also express BTNL2" in the revised manuscript.

Page 6 line 16 ". Notably, there was no significant difference in the number of 16 tumor-infiltrating IL-17A-or IFN- γ -producing CD4⁺T cells, Tregcells, or NK cells"

- should make reference to the model, eg following CT26 challenge s.c.

We thank the reviewer's suggestion, and we added the references which had CT26 tumor challenge in the revised manuscript (pages 6, lines 16-18).

Fig 2c, it should be made clear in the figure legend if the cells have been restimulated in order to induce IL-17 expression.

We thank the reviewer's suggestion, and we clarified it in the figure legend of Figure 2c as "cells were restimulated with Cell Activation Cocktail (with Brefeldin A) for 4 hours" in the revised manuscript.

Page 6 line 20. "Notably, the glycosylation site mutant N4S-BTNL2-Fc promoted significantly greater numbers of $\gamma\delta$ T17 cells and production of IL-17A than did WT BTNL2, either with or without co-stimulation with IL-1 β and IL-23"

- the difference is marginal. Could it be influenced by e.g. higher affinity of deglycosylated protein for the plastic, or slight difference in quantification of the BTNL2-Fc, or slightly higher purity of deglycosylated protein?

We thank the reviewer's questions. We examined the purity of the recombinant WT-BTNL2-FC and N4S-BTNL2-FC proteins by Coomassie Blue Staining, and found that they had comparable purities (**Sup Figure 4a**).

We suspect that the difference may be due to slightly increased affinity between

N4S-BTNL2-Fc and its receptor compared to the WT BTNL2-FC and the receptor. We are trying to identify the receptor of BTNL2 by different methods, and will explore this hypothesis in the future.

What is the tumour model used in fig 2f?

We are sorry not describing it clearly, and the tumor model used in Figure 2f is CT26, and we clarified it in the figure legend of the revised manuscript.

Page 9 line 9 “Importantly, prior studies have classified the LLC tumor model as a non-immunogenic “cold” mouse tumor due to the lack of response to anti-PD-1 therapy.”

-missing a supporting reference

We thank the reviewer’s reminder, and we added two references in the revised manuscript (pages 9, lines 22).

Discussion: “The importance of the $\gamma\delta$ T cell population was further highlighted by a large bioinformatics-based analysis that included thousands of patients and multiple tumor types, which found that the best correlate for overall survival was the presence of tumor-infiltrating $\gamma\delta$ T cells”

- the accuracy of the $\gamma\delta$ gene signature has been challenged by Tosolini et al (doi: 10.1080/2162402X.2017.1284723) so the authors may wish to qualify this statement.

We thank the reviewer’s comments and suggestions. Since there is controversial regarding the correlation between $\gamma\delta$ T cell abundance and the prognosis of the cancer patients, we deleted the sentence of “The importance of the $\gamma\delta$ T cell population was further highlighted by a large bioinformatics-based analysis that included thousands of patients and multiple tumor types, which found that the best correlate for overall survival was the presence of tumor-infiltrating $\gamma\delta$ T cells” and the corresponding reference in the revised manuscript.

Methods

Antibodies used for in vivo depletion, clone names are missing and their source.

We thank the reviewer's comments, and we added the clones' name and the source for the depletion antibodies. Polyclonal antibody usually doesn't have a clone name.

Page 15 line 13. "For the design of the randomization, littermate control mice and BTNL2-KO mice of the same 13 gender were randomly chosen for the in vivo or in vitro experiments."

- what is 'randomly chosen' referring to? Were the mice age and sex-matched in each assay? Was this information recorded and can the authors confirm the sex and age of the mice for each experiment?

We thank the reviewer for the question, and the mice were age and sex matched (we had the genotyping information for each mouse).

We got littermate control mice (BTNL2^{+/-}) and BTNL2 KO mice by breeding BTNL2 KO mice (BTNL2^{-/-}) with BTNL2 heterozygous mice (BTNL2^{+/-}), and sometimes we got more mice than we needed, and in this situation, we randomly chose the age and sex matched control and KO mice for the experiments. For example, we needed 6 pairs of mice for one experiment, and we got 10 pairs of mice at the time, and then we randomly chose 6 pairs of age and sex matched mice out of the 10 pairs.

Reviewer #2 (Remarks to the Author):

Nature Communications manuscript NCOMMS-21-14489

Title: BTNL2 is a suppressor of antitumor immunity by regulating IL-17A-producing $\gamma\delta$ T cells

The data in this manuscript show that butyrophilin like 2 (BTNL2) plays a role in tumor progression by stimulating IL-17A expression in $\gamma\delta$ T cells that encourage neutrophils to suppress CD8 T cells. The authors show that inhibition of BTNL2 through a newly described antibody or genetic deletion of BTNL2 in cancer cells

slows the growth of transplanted cancer cell lines in mice. Using in vitro assays, the authors show that BTNL2 can directly stimulate IL-17A production in $\gamma\delta$ T cells. In mice, decreased IL-17A by inhibition of BTNL2 reduces neutrophils and increases cytotoxic T cells, whose contribution is shown by depleting antibodies. Association of BTNL2 expression in human tumor tissue microarrays and cancer patient survival is shown for lung and colon cancer.

Butyrophilins are an understudied family of proteins, so this manuscript provides novel information on its function in cancer immunology. Overall, this is a nice study with good mechanistic insight. The authors use both loss-of-function and gain-of-function assays to support their conclusions. The experiments include proper controls (except in one case, see below) and statistics are appropriate. I only have a few comments to improve the quality of the manuscript:

We thank the reviewer's comments!

1. The format of the panel figures needs better organization. It is often difficult to find the panels referenced in the text. Sometimes they go down, sometimes they go across. I realize that there is a lot of data, but the panels need to be easy to follow.

We thank the reviewer's suggestion, and we re-organized the panels of the data in the revised manuscript.

2. The MDSC acronym is not defined in Introduction.

We thank the reviewer's reminder, and we defined the "MDSC" in the introduction (pages 3, lines 26).

3. Supp Figure 2f is confusing. According to Supp Figure 1d, the 95 kDa band is non-specific, but point mutations in BTNL2 change the molecular weight of the 95 kDa band. How do the authors reconcile this discrepancy? The authors should show the bands between 55 and 72 kDa.

We thank the reviewer's comment. The data presented in the Sup Figure 2f is

WT-BTNL2-Fc and mutant BTNL2-Fc recombinant proteins (immunoblotted with Fc antibody), and the data in the Sup Figure 1d represents the expression of the endogenous BTNL2 (immunoblotted by using anti-BTNL2 mAb-2 antibody). Since Fc tag has 223 amino acids, so the migration of BTNL2-Fc is slower than the endogenous BTNL2 in the gel.

We provided full uncut gel with protein markers for the **Sup Figure 1d** and **Sup Figure 2f**.

4. In regard to Figure 2, I do not agree with the authors' conclusion of skewed differentiation towards gdT17. The data show that IL-17A is regulated by BTNL2 – they do not show cell differentiation.

We thank the reviewer's comment, and we changed the description of "...both wild-type (WT) BTNL2-Fc and N4S-BTNL2-Fc recombinant proteins were capable of inducing $\gamma\delta$ T17 cell differentiation in splenocytes..." to "...both wild-type (WT) BTNL2-Fc and N4S-BTNL2-Fc recombinant proteins were capable of inducing the production of IL-17A by $\gamma\delta$ T cells in splenocytes..." in the revised manuscript (pages 6, lines 20-22).

5. The sentence describing the data in Figure 2f is incorrect. There is no additive or synergistic effect of anti-IL17A and anti-BTNL2.

We thank the reviewer's comments. The data in Figure 2f indicates that anti-tumor effect of BTNL2 blockage is dependent on IL-17A, as anti-IL-17A neutralization abolishes the anti-tumor effect of BTNL2 blockage. We didn't intend to demonstrate that there is a synergistic effect of anti-IL17A and anti-BTNL2.

6. There is no explanation of the blot in Figure 2g. Please provide description in text.

We thank the reviewer's reminder, and we added the description for the blot of Figure 2g in the revised manuscript (pages 6, lines 31-32).

7. It is surprising to see Vg1 cells producing the majority of IL-17A in the transplant

models, as Vg4 and Vg6 cells usually produce the most IL-17A. Because this is an unusual observation, the authors need to show the full gating strategy for these data with controls for Vg1 and CD27 staining.

We thank the reviewer's suggestion, and we provided the full gating strategy for Figure 3c and 3d. We also provided isotype controls for V γ 1 and CD27 staining in the revised manuscript.

I wonder whether BTNL2 may specifically bind the Vg1 T cell receptor, so the authors may consider this for future work.

We thank the reviewer's suggestion. As the first reviewer also raised this question, we performed the experiment to examine whether BTNL2 directly bind the V γ 1 TCR.

We reconstituted V γ 1 TCR by co-transfecting V γ 1.1TCR and V δ 6.3TCR in 293T cells. We didn't find BTNL2-Fc directly bind V γ 1.1V δ 6.3TCR, and please find the data in the **Sup Figure 4c** of the revised manuscript.

As BTNL2 also binds CD4⁺ T cells and inhibits the activation of CD4⁺ T cells (**Sup Figure 4b**), the receptor for BTNL2 may not be V γ 1 $\gamma\delta$ TCR, as CD4⁺ T cells don't express this molecule.

8. The authors state that neutrophils and MDSCs are different populations (page 7). This is incorrect. There is no way to distinguish a neutrophil from an MDSC by cell surface molecules, because these are in fact the same cells. The authors should be aware that use of the MDSC terminology is being discouraged by experts (see Hegde, Leader & Merad, Immunity 2021) because it is ambiguous. I strongly suggest that the authors remove any mention of MDSC and refer to the CD11b⁺Ly6G⁺ population as neutrophils.

We thank the reviewer's comments and suggestions, and we agree with the reviewer that neutrophils and the granulocytic (CD11b⁺Ly6G⁺Ly6C_{low}) MDSCs have the same surface markers, and basically they are the same cells. We revised the description as "...MDSCs can be further subdivided into two subsets: so-called monocytic (CD11b⁺Ly6G⁺Ly6C_{high}) and granulocytic (CD11b⁺Ly6G⁺Ly6C_{low}) MDSCs, which

have the same surface markers of monocytes and neutrophils”, which is in the pages 7, lines 30-31 of the revised manuscript.

We also revised the manuscript as “Interestingly, almost all of the tumor-infiltrating neutrophils were phenotypically consistent with granulocytic MDSCs, which indicates they are the same cells” in the pages 8, line 4-5 of the revised manuscript.

We agree with the reviewer that the terminology of MDSC is ambiguous and controversial, while the usage of this terminology has historic reason and has been used for many years, and it’s still accepted by many of the researchers in the field. As we already used this terminology throughout the manuscript, and we also used CD11b⁺Gr-1⁺ to mark this population of cells, we feel it may be reasonable to keep the terminology in the manuscript, and we will avoid to use this terminology in the future studies.

REVIEWER COMMENTS

Reviewer #1 (Remarks to the Author):

Thank you for comprehensively addressing the comments that were raised in the point-by-point response. It will be interesting to see what the identity of the BTNL2 ligand is in future studies. I have no additional comments and am satisfied by all these responses.

Reviewer #2 (Remarks to the Author):

The authors have addressed each of comments and suggestions. This is a tremendous amount of work that will greatly add to our knowledge of butyrophilin molecules in cancer. It is a timely publication since a recent study described the role of BTNL2 in shaping gamma delta IELs (Panea et al., Communications Biology 2021). So it seems that BTNL2 suppresses multiple gamma delta T cell subsets. This manuscript adds to a growing field in BTNL biology.

Point-by-point response

We really thank both reviewers' comments and suggestions, which helped us to further improve the quality of this manuscript.

Reviewer #1 (Remarks to the Author):

Thank you for comprehensively addressing the comments that were raised in the point-by-point response. It will be interesting to see what the identity of the BTNL2 ligand is in future studies. I have no additional comments and am satisfied by all these responses.

We thank the reviewer's comments and encourages! We are trying to identify the receptor/ligand of BTNL2 on the T cells.

Reviewer #2 (Remarks to the Author):

The authors have addressed each of comments and suggestions. This is a tremendous amount of work that will greatly add to our knowledge of butyrophilin molecules in cancer. It is a timely publication since a recent study described the role of BTNL2 in shaping gamma delta IELs (Panea et al., Communications Biology 2021). So it seems that BTNL2 suppresses multiple gamma delta T cell subsets. This manuscript adds to a growing field in BTNL biology.

We thank the reviewer's comments and encourages!